# Profiles of cloud condensation nuclei, dust mass concentration, and INP-relevant aerosol properties in the Saharan Air Layer over Barbados from polarization lidar and airborne in situ measurements

Moritz Haarig[1], Adrian Walser[2], Albert Ansmann[1], Maximilian Dollner[2], Dietrich Althausen[1], Daniel Sauer[3], David Farrell[4], and Bernadett Weinzierl[2]

[1]Leibniz Institute for Tropospheric Research (TROPOS), Leipzig, Germany
[2]Faculty of Physics, University of Vienna, Vienna, Austria
[3]Institut für Physik der Atmosphäre, Deutsches Zentrum für Luft- und Raumfahrt (DLR), Oberpfaffenhofen, Germany
[4]Caribbean Institute for Meteorology and Hydrology, Bridgetown, Barbados

*Correspondence to:* Moritz Haarig (haarig@tropos.de)

**Abstract.** The present study aims to evaluate lidar retrievals of cloud-relevant aerosol properties by using polarization lidar and coincident airborne in situ measurements in the Saharan Air Layer over the Barbados region. Vertical profiles of the number concentration of cloud condensation nuclei (CCN), large particles (diameter $d$>500 nm), surface area, and ice nucleating particles (INP) are derived from the lidar measurements and compared with CCN concentrations and the INP-relevant aerosol properties measured in situ with aircraft. The measurements were performed in the framework of the Saharan Aerosol Long-range Transport and Aerosol–Cloud-interaction Experiment (SALTRACE) in summer 2013.

The CCN number concentrations derived from lidar observations were up to a factor of two higher than the ones measured in situ on board the research aircraft Falcon. Possible reasons for the difference are discussed. The number concentration of particles with dry radius >250 nm and the surface area concentration obtained from the lidar observations and used as input for the INP parameterizations agreed well (<30–50% deviation) with the aircraft measurements. In a pronounced lofted dust layer during summer (10 July 2013), the lidar retrieval yielded 100–300 CCN per $cm^3$ at 0.2% water supersaturation and 10–200 INP per L at –25°C.

During the SALTRACE winter campaign (March 2014), the dust layer from Africa was mixed with smoke particles which dominated the CCN number concentration. This example highlights the unique lidar potential to separate smoke and dust contributions to the CCN reservoir and thus to identify the sensitive role of smoke in trade wind cumuli developments over the tropical Atlantic during the winter season.

## 1 Introduction

Climate predictions are highly uncertain (IPCC, 2013). One of the reasons is our poor knowledge of the impact of atmospheric aerosol on cloud processes. To improve our understanding of aerosol-cloud interaction, new techniques for profiling of cloud condensation nuclei (CCN) and ice-nucleating particles (INP) are required. Lidar permits a regular and continuous monitoring of the cloud-relevant aerosol properties up to the tropopause height. Methods have been developed to retrieve CCN and INP-

relevant particle microphysical properties from particle extinction coefficients measured with lidar (Mamouri and Ansmann, 2016; Sawamura et al., 2017; Lv et al., 2018). In the case of INP profiling, particle extinction coefficients are converted to particle number concentrations $n_{250}$ (particles with dry radius >250 nm) and particle surface area concentrations $s$. The $n_{250}$ profile is input in the INP parameterization schemes of DeMott et al. (2010, 2015) and Tobo et al. (2013) and $s$ profiles are in-

5 put in respective INP parameterizations by Niemand et al. (2012), Steinke et al. (2015), Ullrich et al. (2017), McCluskey et al. (2018), and Harrison et al. (2019). The entire lidar-based INP lidar retrieval procedure is described by Mamouri and Ansmann (2016). First comparisons of the CCN lidar retrievals with airborne in situ observations over a polluted Central European site are presented by Düsing et al. (2018). Sawamura et al. (2017) found good agreement of the lidar-derived surface area and volume concentration with coincident airborne in situ observations focusing on air quality and explicitly excluding periods with

10 the presence of dust particles. Airborne INP studies in the Cabo Verde region found around 100 INP per liter at –23°C in the Saharan Air Layer (SAL) (Price et al., 2018). First comparisons of lidar and in situ observations in dusty environments (Eastern Mediterranean) regarding INP can be found in Schrod et al. (2017) and Marinou et al. (2019).

In this article, we present a detailed comparison of ground-based lidar retrievals to airborne in situ observations of CCN number concentration and INP-relevant aerosol properties. Using observations of transported dust over the remote Atlantic 5000 km

west of the source regions in Africa, we demonstrate the capability of the lidar retrievals to predict the aerosol properties relevant to aerosol–cloud interaction. We use the opportunity of the SALTRACE campaign (Saharan Aerosol Long-range Transport and Aerosol–Cloud-interaction Experiment, Weinzierl et al., 2017), conducted in the Caribbean (Barbados region), for this goal. More than 12 weeks of lidar measurements were performed in June–July 2013 (SALTRACE-1), February–March 2014 (SALTRACE-2), and June–July 2014 (SALTRACE-3). A triple-wavelength polarization Raman lidar (Haarig et al., 2017a) of

the Leibniz Institute for Tropospheric Research (TROPOS) was operated at the Caribbean Institute for Meteorology and Hydrology (CIMH), north of Bridgetown, Barbados (13.15°N, 59.62°W, 110 m about sea level). Airborne in situ measurements were performed during SALTRACE-1. An overview of the instrumentation on-board the research aircraft Falcon of the German Aerospace Center (Deutsches Zentrum für Luft- und Raumfahrt – DLR) is given by Weinzierl et al. (2017).

SALTRACE observations of the long-range transported Saharan dust have been published in the SALTRACE special issue

(Groß et al., 2015; Haarig et al., 2017a; Gasteiger et al., 2017; Kandler et al., 2018). The lofted dust plumes in the Saharan air layer (SAL) occur between 1.5–5 km height. Many simultaneous measurements with aircraft and the ground based lidar have been realized during SALTRACE. For our study, we use the Falcon observations of the particle size distribution and of the CCN number concentration. In the lidar-Falcon comparisons, three case studies are analyzed. CCN properties have been studied previously in the Caribbean but without involving vertical profiling with lidar (Siebert et al., 2013; Kristensen et al., 2016;

Wex et al., 2016; Jung et al., 2016). A dust–smoke mixture and a pure marine case from the SALTRACE-2 (winter campaign) are presented in addition to contrast the almost pure dust conditions prevailing during the summer half year. This comparison highlights the strong impact of smoke particles on the CCN levels over the remote tropical Atlantic during the winter half year (biomass burning season).Whereas the pristine marine case demonstrates the aerosol conditions without long-range transport of African aerosol to the Caribbean.

The paper is structured as follows: In Sect. 2, the ground-based and airborne instrumentation and the lidar retrieval are shortly

presented. Then the three Saharan dust cases (used in our comparison study) are described regarding dust layering, the meteorological context, and air mass transport (Sect. 3). In Sect. 4, CCN concentrations and particle number, surface area and mass concentrations obtained from the aircraft and lidar measurements are compared. In Sect. 5, summer and winter lidar observations are contrasted. A summary and concluding remarks are given in the last section.

## 2 Instrumentation and methods

### 2.1 Lidar retrievals of CCN and INP concentrations

The triple-wavelength polarization Raman lidar BERTHA (Backscatter, Extinction, lidar Ratio, Temperature, Humidity profiling Apparatus) described in Haarig et al. (2017a) measures the backscatter coefficient at three wavelengths (355, 532, and 10 1064 nm), the extinction coefficient at 355 and 532 nm and in a new configuration also at 1064 nm (Haarig et al., 2016). The depolarization ratio is measured at 355, 532, and 1064 nm simultaneously. The inelastic (Raman) channels at 387 and 607 nm enabling the independent measurement of the extinction coefficient (Ansmann et al., 1992) can be used at nighttime only. The sunlight causes an enhanced background noise level due to the broad interference filter (3 nm width). In order to have coincident observations of lidar and aircraft, daytime measurements had to be used in the present study. The closest nighttime observation 15 (always from the same date) provided the extinction-to-backscatter ratio (lidar ratio) to constrain the daytime observations. For daytime measurements, the calculation of the backscatter coefficient from the elastic signals at 355, 532, and 1064 nm was performed via the Fernald/Klett method (Klett, 1981; Fernald, 1984).

The conversion from backscatter coefficient and particle linear depolarization ratio (PLDR) to particle number and surface area concentration follows the method described in Mamouri and Ansmann (2015, 2016). The particle depolarization ratio is used to 20 separate the contributions of different aerosol types to the backscatter coefficient: mineral dust (d) with high depolarization ratio (around 0.3), marine aerosol (m) with low depolarization ratio in humid state ($\leq$0.05), and continental aerosol (c) with low depolarization ratio ($\leq$0.05). By multiplication with an appropriate extinction-to-backscatter ratio (lidar ratio, Müller et al., 2007; Groß et al., 2013; Baars et al., 2016) the backscatter coefficients are converted into extinction coefficients (see also description in Marinou et al., 2019). These extinction coefficients were verified independently by BERTHA's Raman lidar measurements 25 at nighttime. Long-term AERONET observations (columnar particle number concentrations and aerosol optical depth (AOD), Holben et al., 1998) are used to derive empirical conversion factors from extinction coefficients to particle number and surface area concentrations (Mamouri and Ansmann, 2016; Ansmann et al., 2019). The respective equations and the conversion factors are listed in Table 1. The AERONET data are filtered for dust events (Ångström exponent AE<0.3, AOT>0.1 at 500 nm), pure marine (0.25<AE<0.6, AOT<0.07) and continental (AE>1.6) conditions. It should be mentioned that the conversion factor 30 for small continental aerosol particles ($n_{50,c}$, particles with radius >50 nm), is obtained using AERONET data from Leipzig (Central Europe), but with a factor of 0.5 to best approximate the African rural aerosol conditions (Shinozuka et al., 2015).

In a next step, INP and CCN concentrations are retrieved. INP parameterizations have been developed for the aerosol types dust, soot, and marine particles (see Table 2). The number concentration $n_{250}$ and the surface area concentration $s$ are the aerosol-

relevant input parameters which are obtained by conversion of the lidar-derived particle extinction profiles. In the present study, we focus on immersion freezing, i.e., ice nucleation by an INP immersed into a liquid-water droplet. The parameterization by DeMott et al. (2010) is used for the dust and non-dust (continental or marine) particles with radius >250 nm, whereas the DeMott et al. (2015) parameterization is explicitly developed for dust particles. Harrison et al. (2019) developed a parameter-

5 ization for the very ice-active mineral K-feldspar which is part of the Saharan dust. Mineralogical measurements at Barbados show that only 1% of the dust particles consists of K-feldspar (Kandler et al., 2018). Therefore, we assume the K-feldspar parameterization by Harrison et al. (2019) is valid for 1% of the total surface area of dust. The surface-area based INP param-eterization developed by Ullrich et al. (2017) leads to much higher values and are not shown in this study. McCluskey et al. (2018) developed a parameterization for marine aerosol with samples from the Atlantic Ocean. The INP parameterizations and

10 input parameters are listed in Table 2.

To estimate the CCN number concentration $n_{CCN}$, Mamouri and Ansmann (2016) use a dry activation diameter of 200 nm for dust and 100 nm for continental pollution and marine particles at 0.15–0.2% water supersaturation. An enhancement factor $f_{ss}$ determined in Mamouri and Ansmann (2016) from various laboratory and field studies (activation diameter and supersatura-tion) is used to retrieve $n_{CCN}$ for different supersaturation levels (Table 1). The supersaturation of 0.2% with respect to water

is motivated by the findings of Wex et al. (2016) who reported this as a typical value for trade wind cumuli in the Barbados region. CCN concentrations at the same supersaturation (0.2%) were measured in situ with a CCN counter aboard the Falcon research aircraft.

The use of the different activation diameters (100 nm for continental pollution aerosol and for marine particles, 200 nm for dust) is motivated by the following facts. Based on kappa-Köhler theory (Petters and Kreidenweis, 2007), we computed the

20 activation diameter for 0.2% water supersaturation and temperatures from –10°to 20°C for various materials and chemical compositions (Table 3), in which 10°C is the most realistic value within the SAL as indicated by local radiosondes. Fresh Saharan dust mimicked by dry-generated dust is very hydrophobic (low hygroscopicity parameter $\kappa$) so that the activation diameter is around 275 nm (at 10°C). Cloud-processed Saharan dust particles (mimicked by wet-generated dust samples) may have changed their hygroscopic properties (higher $\kappa$ value) so that their CCN efficacy increased. Laboratory studies with

25 wet-generated dust particles (in contrast to dry-generated fresh dust particles) reported higher $\kappa$ values (Koehler et al., 2009; Herich et al., 2009; Kumar et al., 2011b). However, although the Saharan dust was transported over several thousands of kilo-meters across the Atlantic Ocean, observations suggest that the dust in the SAL remained nearly unprocessed (Lieke et al., 2011; Denjean et al., 2015; Weinzierl et al., 2017; Kandler et al., 2018). Therefore, $\kappa$ should not change significantly dur-ing transport and be closer to the value for fresh Saharan dust which is taken from laboratory studies (Koehler et al., 2009;

Herich et al., 2009; Kumar et al., 2011a, b). Twohy et al. (2009) found good agreement for their CCN measurements with a $\kappa$ of 0.05 in the Eastern Atlantic. Herich et al. (2009) concluded that the activation diameter for Saharan dust (dry generated) is most probably 200 nm at a supersaturation of 0.2%. This is confirmed by studies of Shinozuka et al. (2015) and Lv et al. (2018). Following this discussion, we assume an activation diameter of 200 nm for Saharan dust at Barbados which corre-sponds to a $\kappa$ value of approximately 0.05.

The activation diameter for continental aerosol particles (fine-mode pollution) depends on their chemical composition. Kandler et al.

(2018) found sulfate particles as a dominant contribution of continental pollution aerosol in the SAL, but the instrumentation was not suitable to detect organics. Considering ammonium sulfate with a small contribution of less hydrophilic organic particles as continental aerosol within the SAL, a dry activation diameter of 100 nm at a supersaturation of 0.2% is a suitable estimate and therefore used in this study.

We assume sea salt to be the dominant component of the marine aerosol, and prescribe an activation diameter of 70 nm (Table 3, at 0.2% supersaturation and 10°C). Whereas Mamouri and Ansmann (2016) estimated a dry activation diameter of 100 nm based on literature. Going from 100 nm to 70 nm as activation diameter, would increase $n_{CCN}$ by a factor of approximately 1.5.

In conclusion, we used a dry activation diameter of 200 nm for Saharan dust, and of 100 nm for continental and marine parti-
cles, assuming a supersaturation of 0.2% in the SALTRACE studies.

The polarization lidar–photometer networking technique (POLIPHON) introduced by Mamouri and Ansmann (2014, 2017) delivers mass concentrations of fine and coarse mode dust, i.e, dust particles with diameter $d<1$ $\mu$m and $d>1$ $\mu$m, respectively (Table 1). The PLDR at 532 nm is used to separate the contributions of non-dust aerosol (PLDR=0.05), fine-mode dust (PLDR=0.16), and coarse-mode dust (PLDR=0.35). The lidar-derived extinction coefficient of the fine and coarse mode dust
component is converted into volume concentration using long-term AERONET data sets and finally to mass concentration using the mass density of dust (2.6 g/cm$^3$) (Mamouri and Ansmann, 2014, 2017).

## 2.2   Airborne in situ aerosol measurements

A full list and details of the instrumentation installed aboard the research aircraft Falcon of the DLR are given in Weinzierl et al. (2017). Information on size-resolved particle number concentrations are obtained from condensation particle counters and opti-
20  cal particle spectrometers. The condensation particle counters were operated at slightly different cutoff diameters around 10 nm. The spectrometer setup included an airborne version of the Ultra High Sensitivity Aerosol Spectrometer (Cai et al., 2008; Brock et al., 2011; Kupc et al., 2018), a Grimm model 1.129 SkyOPC, and a Cloud and Aerosol Spectrometer (Baumgardner et al., 2001). The combination of these spectrometers covers the complete range of particle diameters from about 70 nm to 50 $\mu$m. Particle number size distributions (NSDs) are derived from the entirety of these data using a consistent Bayesian inversion
method (Walser et al., 2017). Here, the NSDs are approximated by trimodal log-normal distributions. In situ cloud condensation nuclei concentrations are measured with a Cloud Condensation Nuclei Counter (Roberts and Nenes, 2005; Lance et al., 2006) operated at a water vapor supersaturation of 0.2%. These concentrations are corrected for losses of large CCN at the aircraft's isokinetic aerosol inlet.

## 3   Lidar observations of SAL dust layering: Comparison days

Three cases of the SALTRACE summer-2013 campaign were selected for in-depth comparisons of lidar and aircraft observations: 22 June, 10 July, and 11 July 2013. The criteria for the selection were based on the low spatial distance between the lidar site and the Falcon aircraft (flight patterns in the Barbados region, see Fig. 1). The time-height displays of the volume

depolarization ratio at 532 nm shown in Fig. 2 indicate very homogeneous dust structures in the SAL on these three days and thus good conditions for comparisons. Daytime lidar observations are used to have coincident measurements with the Falcon aircraft. Below 2 km height, trade wind cumuli attenuated the lidar signals. Only the cloud-free profiles were used to calculate the mean backscatter coefficient and depolarization ratio. Table 4 contains information about the measurement periods of the

Falcon aircraft and the lidar including the mean horizontal distance of the Falcon from the lidar site and flight height levels. Except for two flight legs, the mean distance was below 100 km. In the SAL, winds from eastward directions with a wind speed between 10 and 18 m/s prevailed leading to a dust transport of 35–65 km/h. The lidar profiles were averaged over 100–140 minutes which corresponds to a spacial average of 60–150 km considering the wind speed. Therefore, the Falcon aircraft and the ground-based lidar observed in principle the same dust layer at these selected days.

A weak dust outbreak was observed on 22 June 2013 (Fig. 2a–b), belonging to the first out of four main dust periods during SALTRACE-1 (Groß et al., 2015). The trajectories (not shown) indicate a possible dust uptake over Mali and Mauritania 8–9 days prior to the arrival at Barbados. In contrast to the later two cases, these air masses spent more time in the populated coastal region of west Africa (Senegal) and so the probability of anthropogenic influence was high.

After the passage of the tropical storm Chantal (Weinzierl et al., 2017), a strong and stable flow of Saharan dust towards the

Caribbean established and lasted for more than 4 days (10–13 July 2013). We use the 10 and 11 July observations for the comparison study. The Saharan Air Layer (SAL) extended vertically from 1.8 km to almost 5 km height as shown in Fig. 2c-f. As already discussed in Haarig et al. (2017a) based on backward trajectory analysis and the particle depolarization ratio measurements, pure dust conditions (with rather low probability of contamination with anthropogenic pollution) were given. The dust traveled 5–7 days over the Atlantic Ocean.

The CCN and INP parameterizations are aerosol-type dependent. Therefore, a separation into a dust and non-dust (continental or marine aerosol) is necessary as done in Mamouri and Ansmann (2016) and Marinou et al. (2019). Pure Saharan dust has a PLDR at 532 nm of 0.31±0.03 (Freudenthaler et al., 2009), whereas continental pollution / smoke and marine aerosol have a PLDR ≤0.05 (Groß et al., 2013; Baars et al., 2016). The particle depolarization ratio is the best indicator for the presence of dust. Its vertical profile (Fig. 2) indicates that not only dust was transported in the SAL, but a mixture of dust and non-

dust. On 10 and 11 July 2013, however, only a rather small non-dust component was present (layer mean PLDR at 532 nm of 0.29±0.02 and 0.31±0.02, respectively). In contrast, on 22 June 2013 the non-dust component was significant (PLDR of 0.25±0.03). The indicated uncertainty considers systematic errors and statistic uncertainties in the lidar data analysis. Because of the geographical location of Barbados, backward trajectories were not sufficient to decide whether the non-dust component was of marine or continental origin. Instead the method described in Ansmann et al. (2017) was applied which uses the fact

that continental aerosol particles have a significantly higher lidar ratio (50 sr) due to considerable light absorption and much smaller particle sizes than the ones of marine aerosol particles (20 sr). The independently measured total particle extinction coefficient from our Raman lidar measurements (Ansmann et al., 1992) is compared to the sum of the extinction coefficients obtained by multiplying the type-separated backscatter coefficients with the respective type-dependent lidar ratios. An example will be shown in Section 5. A good agreement was found for continental pollution aerosol in the SAL (> 2 km height) and

marine aerosol in the marine aerosol layer below (<2 km height). Often Raman lidar observations could not be performed at

bright daylight conditions. In these cases, we had to use Raman lidar measurements after sunset to check the non-dust aerosol type in the SAL.

## 4 Lidar retrievals versus airborne in situ aerosol observations

We begin with comparisons of CCN concentrations ($n_{\mathrm{CCN}}$) in Sect. 4.1. Particle number concentrations $n_{250}$ of large particles

and surface area concentrations $s$ are then compared in Sect. 4.2. In Sect. 4.3, we show simultaneous observed profiles of fine mode and coarse mode mass concentrations.

### 4.1 CCN profiles

In Figure 3, the lidar-derived number concentration of CCN for dust $n_{\mathrm{CCN},d}$ (red line) and continental pollution particles $n_{\mathrm{CCN},c}$ (olive line) are presented. The total CCN number concentration $n_{\mathrm{CCN}}$ (black line, lidar) can be compared with measurements

of the cloud condensation nuclei counter on board the Falcon aircraft (black dots) at the same supersaturation. In Table 5, the vertically averaged values are compared. The lidar-derived $n_{\mathrm{CCN}}$ values are up to twice as large as the in situ measured values. However, the lidar retrieval uncertainty is quite large (factor 2–3). The retrieval uncertainty results from the uncertainty in determining the extinction-to-number-concentration conversion factor for small particles (r$\geq$50 nm or r$\geq$100 nm) using AERONET derived AOD and columnar number concentrations ($n_{50}, n_{100}$) as described in Mamouri and Ansmann (2016). Besides the

large retrieval uncertainty, other uncertainty sources may have contributed to the systematic bias between the lidar and airborne in situ observations: (i) The lidar conversion factors are derived for AERONET stations close to the Sahara. These conversion factors may not be applicable to aged dust after long-range transport, and may overestimate the occurring accumulation mode dust particle number concentration and thus $n_{100,d}$. (ii) The used dust activation diameter ($d_{\mathrm{dry}}$=200 nm) may have been too low and the true one was much larger than 200 nm (see Table 3, $d_{\mathrm{dry}}$=275 nm for dry-generated (fresh) dust) and thus less dust

particles were activated in the cloud condensation nuclei counter aboard the Falcon than estimated by lidar. (iii) Horizontal and temporal inhomogeneities in the dust concentration along the flight tracks and over the lidar site may have also contributed to the found differences. (iv) Although the Falcon data are corrected for the particles losses at the inlets (Spanu et al., 2019), there are several uncertainty sources in the in situ CCN measurement, that may have contributed to the found bias.

Overall, the CCN number concentration for the three presented dust cases agrees within a factor of two between the in situ mea-

surement and the lidar retrieval. As the behavior is the same for all three comparison studies, it is expected to be representative for Saharan dust episodes in the Caribbean.

### 4.2 INP-relevant aerosol profiles

In Figure 4a–c, the profiles of the sum of $n_{250,d}$ and $n_{250,c}$ are compared with the integral values of the particles number size distribution for $r_{\mathrm{dry}}$>250 nm measured on board the Falcon aircraft. The in situ values are transformed to the pressure and

temperature at the measurement altitude to be comparable with the lidar observations. As can be seen, the in situ and lidar values agree well, except on 22 June and 11 July in the lower part of the SAL, where horizontal inhomogeneities in the dust

load (see Fig. 2) may have partly caused the differences between the two measurements. The contribution of continental smoke and pollution aerosol to $n_{250}$ was less than 3% in the SAL during the strong dust outbreak on 10–11 July 2013 and about 10% on 22 June 2013. In total, there were less than 40 particles ($r_{dry}$>250 nm) per $cm^3$ in all three cases over the remote Atlantic. Fig. 4d–f compares the profiles of the total surface area concentration derived from lidar extinction coefficients and from the airborne in situ measured number size distribution. Here, the contribution of the continental pollution particles to $s$ within the SAL is 4–6% during the strong dust outbreak (10–11 July) and 20% on 22 June 2013. The lidar values are considerably larger than the in situ values. The use of too large conversion factors (based on AERONET observations close to the Sahara) may be one of the reasons for the strong disagreement.

An example on INP profiling is given in Fig. 4g–h at a temperature of –25°C. The DeMott et al. (2010) and DeMott et al. (2015) parameterization (including the correction factor of 3) is used with $n_{250,d} + n_{250,c}$ and $n_{250,d}$ profiles as input, respectively. Furthermore, the Harrison et al. (2019) parameterization for K-feldspar was added with a 1% contribution of K-feldspar to the dust surface area concentration as indicated by Kandler et al. (2018). The uncertainty range (factor 3) is exemplarily indicated for the DeMott et al. (2015) parameterization by the dashed line. As can be seen, the SAL contains INP concentrations of 10–200 $L^{-1}$ at –25°C.

## 4.3  Fine and coarse mode mass concentrations

As an additional feature to the CCN and INP profiles, the dust mass concentration can be derived from the lidar measurements separately for fine and coarse mode dust (Table 1). The comparison with airborne in situ observations are shown in Fig. 5. The mass concentrations are calculated from the lidar derived and in situ measured volume concentration by assuming a dust mass density of 2.6 $g/cm^3$. An excellent agreement is obtained for the coarse mode. This indicates that the Falcon measurements capture well the large particles in the SAL (Spanu et al., 2019). The coarse-mode mass concentration from POLIPHON is around 16 times higher than the fine-mode mass concentration leading to a mass fine-mode fraction of 0.06. For the optical properties, such as the backscatter coefficient, the fine-mode fraction is 0.2. These mass (or volume) and backscatter fractions are in full agreement with simultaneous AERONET sun photometer observations of the fine-mode volume and AOD fractions at Ragged Point, Barbados. Again, for the fine particle dominated quantities, i.e., the fine-mode mass concentration, the lidar derives higher values than observed in situ. Uncertainties in in situ aerosol measurements or in the lidar conversion factors might be the reason. However, a good agreement of the lidar products with AERONET observations is found and corroborates the quality of the lidar products.

## 5  Contrasting pure dust with mixed dust–smoke and pristine marine conditions

We use the opportunity of SALTRACE to contrast the presented dust dominated cases during summer (SALTRACE-1) with a pristine marine measurement and a dust–smoke mixture during the SALTRACE-2 winter campaign (Haarig et al., 2017b). No aircraft measurements are available for SALTRACE-2.

## 5.1 Pristine marine conditions

Caribbean background cases without aerosol transport from Africa were found during SALTRACE-2. End of February 2014, pristine marine conditions prevailed at Barbados as already discussed in Haarig et al. (2017b). The 26 February 2014 was chosen for the present study as the influence of dry marine particles (Haarig et al., 2017b) was less pronounced as the days before (23 and 24 February 2014). The results are presented in Fig. 6. The marine aerosol reached 2 km height (Fig. 6a). The low values of PLDR ($\leq$0.03) shown in Fig. 6b increased at the top of the marine aerosol layer to values of 0.06 indicating the presence of dry marine particles with a non-spherical shape as discussed in Haarig et al. (2017b). These particles are misclassified as a very small dust contribution as can be seen in Fig. 6c for the CCN number concentration. Otherwise the CCN reservoir consists of marine aerosol only (up to 250 per cm$^3$). The dashed line in Fig. 6c indicates the lidar-retrieved $n_{\text{CCN}}$ from 3 March 2014 showing a similar behavior. The INP reservoir at –25°C (Fig. 6d) derived with the parameterization of McCluskey et al. (2018) consists of 0.002–0.1 INP per L, which is around 3 orders of magnitude lower than in presence of Saharan dust. These findings from a remote sensing perspective show the influence of Saharan dust on the cloud properties in the Caribbean and are corroborated by previous helicopter-based in situ measurements in the framework of the CARRIBA project (Cloud, Aerosol, Radiation and tuRbulence in the trade wInd regime over BArbados, Siebert et al., 2013). The marine background without aerosol long-range transport from Africa may be representative throughout the year for the marine contribution in the marine aerosol layer (below the temperature inversion at around 1.5–2.0 km). If dust is present in the SAL above, especially in summer, dust particles are mixed downwards in the marine aerosol layer and add to the marine (background) particles, significantly influencing the CCN and INP reservoir.

## 5.2 Dust–smoke mixture

A pronounced outbreak of aerosol from Africa reached Barbados in the beginning of March 2014. The trajectories ending at 2000 m above ground level on 3 March 2014 (not shown) point to the Sahara as dust source and West Africa (Senegal, Guinea) regarding the source region for biomass burning smoke. The transport across the Atlantic Ocean took around 2 weeks. We use the opportunity of the dust–smoke aerosol mixtures to highlight the strong impact of smoke on the CCN conditions. The transport of biomass burning smoke from Africa towards South America and the Caribbean during wintertime has been previously reported (Ansmann et al., 2009; Baars et al., 2011). An indication for the strong smoke contribution to the measured backscatter signal was the relatively low particle depolarization ratio ($\leq$0.17). Fine-mode smoke does not depolarize laser light (PLDR $\leq$0.05). Figure 7 gives an overview of the measurements on 3 March 2014. A lofted layer (1.6–3.1 km height) of dust and smoke was found above the marine aerosol layer reaching to 1.6 km height. The vertical profiles in Fig. 7b and c show mean values for the time interval from 22:30 to 23:20 UTC. The particle backscatter coefficient (Fig. 7b) is separated into a dust component and a non-dust component using the PLDR separation technique as described in Sect. 2. To estimate whether the non-dust component is of marine or continental origin, the extinction coefficient was calculated from the different contributions to the backscatter coefficient as previously described in Sect. 2 and in Ansmann et al. (2017). The dust-related backscatter coefficient was multiplied by the dust lidar ratio ($S_d$=55 sr), and the non-dust backscatter coefficient by the lidar

ratio for marine particles ($S_m$=20 sr, contributing to the blue curve in Fig. 7c) and for continental pollution particles ($S_c$=50 sr, contributing to the green curve in Fig. 7c). The sum of the extinction coefficient (dust + marine and dust + continental) is then compared with the total extinction coefficient (black curve in Fig. 7c) derived independently with the Raman lidar method (Ansmann et al., 1992). As can be seen, the lofted aerosol layer obviously contains a mixture of dust and smoke, whereas the layer below is dominated by marine particles.

In the next step, $n_{100,d}$, $n_{50,c}$, and $n_{50,m}$ (Fig. 7d) are computed, and the resulting $n_{CCN}$ (Fig. 7e) at 0.2% supersaturation is calculated. The continental pollution contribution to the CCN number concentration is 4 times stronger than the one from the dust aerosol. Thus, in the winter half year with significant smoke contribution from Africa, rather different CCN conditions are found across the Atlantic, leading to likely changes in trade wind cumulus cloud microphysical properties compared to the summer months when dust particles are dominating the CCN reservoir.

In contrast, $n_{250}$ is dominated by mineral dust (Fig. 7f). The lidar-derived contributions to the surface area concentration (Fig. 7g) of dust and smoke are equal. The INP concentration at –25°C estimated in Fig. 7h shows a weak contribution of marine particles (McCluskey et al., 2018) with 3–5 orders of magnitude less efficiency than the dust particles in the lofted layer. The immersion-freezing INP parameterizations based on $n_{250}$ (DeMott et al., 2010, 2015) lead to values around 10 L$^{-1}$. The results are added in Table 5.

Fig. 8 highlights the sensitive impact of smoke aerosol on the CCN concentration. The dust contribution to the optical properties (Fig. 8a) is almost 100% in summer during strong dust outbreaks and around 50% during the biomass burning season, which is in full agreement with AERONET observations. Dust dominates the aerosol mass concentration in the SAL (Fig. 8b) throughout the year, disregarding summer or winter conditions. In strong contrast, the smoke CCN concentration (Fig. 8c) derived with lidar strongly varies between summer and winter. CCN levels of 200–300 cm$^{-3}$ are derived during the strong dust outbreaks in summer (dust contribution around 80%), but close to 500 cm$^{-3}$ in the March 2014 event with a strong contribution of smoke particles (80%).

## 6 Summary and conclusion

For the first time, we compared lidar-derived concentrations of CCN, particle number ($n_{250}$) and total surface area concentration, and fine-mode and coarse-mode dust mass concentration with airborne in situ measurements in vertically deep plumes of aged mineral dust. The study was based on observations in the Saharan air layer over Barbados in the Caribbean, more than 5000 km west of the dust source in Africa. We found good agreement in the case of mass concentrations and large particle number concentrations ($n_{250}$) which serves as input in the INP parameterizations of DeMott et al. (2010, 2015). Differences were observed regarding CCN concentrations. The reason for the differences could not be easily reconciled because many error sources can potentially contribute to the overall uncertainty. The assumptions in the lidar retrieval lead, e.g., to an uncertainty range within a factor of two. Considering this uncertainty, the agreement with the airborne CCN in situ observations is good. We applied several INP parameterization schemes available in literature. The range of solutions provided insight into the uncertainties in the lidar-based INP retrieval. In the case of fine-mode and coarse-mode mass concentrations an excellent

agreement between the lidar and the in situ observations was obtained. This agreement demonstrates the capability of the airborne measurements to capture the large particles as well as the ability to derive accurate dust mass concentrations from lidar observations.

The dominating contribution of smoke particles to the CCN concentration in the wintertime SAL was demonstrated. Furthermore, a lidar observation during pure marine background conditions over Barbados in winter was discussed. At marine conditions the INP concentration is about 3 orders of magnitude lower than during dusty conditions.

As an outlook, further comparisons of lidar and in situ airborne observations of aerosol microphysical properties, CCN and INP concentrations are required predominantly in complex aerosol mixtures of mineral dust and anthropogenic pollution to confirm the robustness of the lidar retrieval and the usefulness of the lidar products. We tested the lidar method for dust (and thus the coarse-mode dominated dust conversion factors) but in the next step we need to extend the studies towards complex aerosol mixtures including fine-mode dominated aerosol types such as smoke and urban pollution. Fine mode conversion factors are very different from the ones for dust. Our validation effort will be continued in the Eastern Mediterranean, where complex mixtures of anthropogenic haze, Middle Eastern and Saharan dust (partly aged and polluted, partly freshly emitted) are present. The simultaneous observations of a lidar in Limassol, Cyprus, and the Falcon aircraft performed in the framework of the A-LIFE campaign (Absorbing aerosol layers in a changing climate: aging, lifetime and dynamics) in April 2017 will be used for this study.

Once, the lidar retrievals are validated and refined by airborne in situ observations, cloud-relevant aerosol properties such as $n_{CCN}$ and $n_{INP}$ can be monitored with organized lidar networks such as the European Aerosol Research Lidar Network (EARLINET, Pappalardo et al., 2014), or continuously operating lidar systems in the framework of PollyNET (Baars et al., 2016). Furthermore, global lidar observations, e.g., from space with CALIPSO (Winker et al., 2009) and in future with EarthCARE (Illingworth et al., 2015) will benefit from the well tested CCN and INP lidar retrievals. Such datasets are needed for improved aerosol-cloud interaction studies and as input in weather and future climate predictions to better consider aerosol particles in respective modeling efforts.

**Author contribution**

MH analyzed the lidar data, and performed together with DA and AA the lidar measurements. AW, together with MD, DS and BW performed the in situ measurements and analyzed the data. MH prepared the manuscript in close cooperation with AA and helpful comments and discussions of AW and BW. DF enabled the lidar measurements at the CIMH, Barbados.

**Competing interests**

The authors declare that they have no conflict of interest.

**Data availability**

The lidar observations (level 0 data, measured signals) and the analysis products are available at TROPOS upon request (info@tropos.de). The aircraft data are available via the University of Vienna upon request.

*Acknowledgements.* This activity is supported by ACTRIS Research Infrastructure (EU H2020-R&I) under grant agreement number 654109
5   and by the European Research Council under the European Community's Horizon 2020 research and innovation framework program/ERC grant agreement number 640458 (A-LIFE). The logistical support of the Caribbean Institute for Meteorology and Hydrology (CIMH), Husbands, Barbados, should be acknowledged. We thank the two anonymous reviewer for their time and effort put into the improvement of the manuscript. We especially thank Paul DeMott for the helpful comments and clarification to the parameterizations developed by his group.

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

**Table 1.** List of abbreviations, formulas and uncertainties for the lidar-derived input parameter to estimate CCN and INP number concentrations (Mamouri and Ansmann, 2016) and for the separation of fine and coarse mode mass concentration (Mamouri and Ansmann, 2017). For Saharan dust the updated conversion factors of Ansmann et al. (2019) are used. All conversion factors are given for a lidar wavelength of 532 nm. In the following, the indices $d$, $c$, and $m$ represent the aerosol types dust ($df$ – fine mode ($r<500$ nm), $dc$ – coarse mode dust ($r>500$ nm)), continental and marine particles, respectively. The extinction coefficient is calculated as the product of the lidar ratio $S_i$ ($S_d=55$ sr, $S_c=50$ sr, $S_m=20$ sr) and the backscatter coefficient $\beta_i$ of the aerosol component $i$. NC and MC stand for particle number concentration and mass concentration, respectively. The density $\rho_d$ of dust is 2.6 g/cm$^3$.

| Symbol | Name | Formula | Unit | Uncertainty |
|---|---|---|---|---|
| $M_{df}$ | fine mode dust MC ($r<500$ nm) | $= \rho_d\, c_{v,df}(S_d\beta_{df})$ with $c_{v,df} = 0.22\ 10^{-12}\text{Mm}$ | $\mu$g m$^{-3}$ | 40–60% |
| $M_{dc}$ | coarse mode dust MC ($r>500$ nm) | $= \rho_d\, c_{v,dc}(S_d\beta_{dc})$ with $c_{v,dc} = 0.8\ 10^{-12}\text{Mm}$ | $\mu$g m$^{-3}$ | 25–35% |
| $n_{50,c}$ | NC with $r_{dry}>50$ nm (cont.) | $= c_{60,c}\ (S_c\beta_c)^{\chi_d}$ with $c_{60,c} = 12.7$ cm$^{-3\,*}, \chi_c = 0.94$ | cm$^{-3}$ | Factor of 2 |
| $n_{50,m}$ | NC with $r_{dry}>50$ nm (marine) | $= c_{100,m}\ (S_m\beta_m)^{\chi_m}$ with $c_{100,m} = 7.2$ cm$^{-3\,*}, \chi_m = 0.85$ | cm$^{-3}$ | Factor of 2 |
| $n_{100,d}$ | NC with $r_{dry}>100$ nm (dust) | $= c_{100,d}\ (S_d\beta_d)^{\chi_d}$ with $c_{100,d} = 4.12$ cm$^{-3\,*}, \chi_d = 0.83$ | cm$^{-3}$ | Factor of 2 |
| $n_{250}$ | NC with $r_{dry}>250$ nm | $= c_{250,i}(S_i\beta_i)$ with $c_{250,d} = 0.19$ Mm cm$^{-3}$ $c_{290,c} = 0.10$ Mm cm$^{-3}$ $c_{500,m} = 0.06$ Mm cm$^{-3}$ | cm$^{-3}$ | 30% |
| $s$ | surface area concentration | $= c_{s,i}(S_i\beta_i)$ with $c_{s,d} = 2.4$ Mm $\mu$m$^2$ cm$^{-3}$ $c_{s,c} = 2.8$ Mm $\mu$m$^2$ cm$^{-3}$ $c_{s,m} = 0.63$ Mm $\mu$m$^2$ cm$^{-3}$ | $\mu$m$^2$ cm$^{-3}$ | 30–50% |
| $n_{CCN}$ | NC of CCN | $= f_{ss,d}n_{100,d} + f_{ss,c}n_{50,c} + f_{ss,m}n_{50,m}$ with $f_{0.2\%,i} = 1.0$ | cm$^{-3}$ | Factor of 2 |
| $n_{INP}$ | NC of INP | see literature in Table 2 | L$^{-1}$ | Factor of 3 |

* for an extinction coefficient of 1 Mm$^{-1}$

**Table 2.** The INP parameterizations for immersion freezing with their references and valid temperature intervals. In the case of immersion freezing, ice nucleation starts via an INP immersed into a liquid droplet.

| | Reference | Temp. (K) | Input | Comments |
|---|---|---|---|---|
| D10 | DeMott et al. (2010) | 238 – 264 | $n_{250,c}, T$ | all aerosol |
| D15d | DeMott et al. (2015) | 238 – 252 | $n_{250,d}, T$ | dust |
| H19d | Harrison et al. (2019) | 235.5 – 269.5 | $s_d, T$ | dust, K-Feldspar |
| U17d | Ullrich et al. (2017) | 243 – 259 | $s_d, T$ | dust |
| U17c | Ullrich et al. (2017) | 237 – 255 | $s_c, T$ | soot |
| M18m | McCluskey et al. (2018) | 245 – 263 | $s_m, T$ | marine aerosol |

**Table 3.** Dry activation diameter $d_{\mathrm{act}}$ for various chemical compositions calculated with kappa-Köhler theory (Petters and Kreidenweis, 2007). The $\kappa$ values are estimated from literature (Ko09 – Koehler et al. (2009), He09 – Herich et al. (2009), Ku11a – Kumar et al. (2011a), Ku11b – Kumar et al. (2011b), Pe&Kr07 – Petters and Kreidenweis (2007), Pe09 – Petters et al. (2009), Kr12 – Kristensen et al. (2012)). The uncertainty in $\kappa$ can be considerable, especially for Saharan dust and organics.

| Material | $\kappa$ | Reference | $d_{\mathrm{act}}(-10^{\circ}\mathrm{C})$ | $d_{\mathrm{act}}(0^{\circ}\mathrm{C})$ | $d_{\mathrm{act}}(10^{\circ}\mathrm{C})$ | $d_{\mathrm{act}}(20^{\circ}\mathrm{C})$ |
|---|---|---|---|---|---|---|
| | | | nm | nm | nm | nm |
| Dry-generated Saharan dust | 0.02 | Ko09, He09, Ku11a, Ku11b | 307 | 290 | 275 | 261 |
| Wet-generated Saharan dust | 0.30 | Ko09, He09, Ku11a, Ku11b | 126 | 119 | 113 | 107 |
| Ammonium sulfate | 0.61 | Pe&Kr07 | 100 | 94 | 89 | 85 |
| Ammonium nitrate | 0.67 | Pe&Kr07 | 97 | 91 | 87 | 82 |
| Low-hygroscopic organics | 0.05 | Pe&Kr07, Pe09, Kr12 | 228 | 216 | 204 | 194 |
| Hygroscopic organics | 0.30 | Pe&Kr07, Pe09, Kr12 | 126 | 119 | 113 | 107 |
| Sodium chloride | 1.28 | Pe&Kr07 | 78 | 74 | 70 | 66 |

**Table 4.** Lidar and Falcon aircraft measurement periods. The mean distance (with standard deviation) of the Falcon from the lidar observation site is given. Local radiosonde launches provide the wind direction (WD) and wind speed (WS) at the altitude of Falcon aircraft.

| Date | Falcon observation | | Lidar observation | Distance | WD | WS |
|---|---|---|---|---|---|---|
| | Height asl.(m) | Time (UTC) | Time (UTC) | km | ° | m/s |
| 22 June 2013 | 2238 | 20:11–20:50 | 19:28–21:20 | 91±60 | 113 | 9.9 |
| | 3369 | 19:28–20:08 | 19:28–21:20 | 94±62 | 51 | 1.0 |
| 10 July 2013 | 2594 | 16:46–16:55 | 17:01–19:25 | 130±100* | 100 | 18.0 |
| | 3560 | 18:12–18:21 | 17:01–19:25 | 20±7 | 93 | 17.9 |
| | 4204 | 17:52–18:10 | 17:01–19:25 | 66±45 | 89 | 14.5 |
| | 4369 | 16:30–16:40 | 17:01–19:25 | 220±2 | 93 | 13.8 |
| 11 July 2013 | 2102 | 14:02–14:13 | 12:40–14:20 | 38±7 | 73 | 12.8 |
| | 2590 | 13:51–14:01 | 12:40–14:20 | 22±13 | 71 | 12.8 |
| | 4196 | 13:39–13:47 | 12:40–14:20 | 17±11 | 64 | 14.7 |

* consists of two measurement periods, one around 220 km away (16:46–16:55 UTC) and one around 30 km away (18:23–18:32 UTC)

**Table 5.** Layer mean CCN and INP concentrations ($n_{CCN}$, $n_{INP}$) in the upper (>3 km) and lower (2–3 km) part of the SAL from lidar and Falcon ($n_{CCN}$ only). The standard deviation of the layer mean is given. The uncertainty range for the lidar retrieval is a factor of 2 for $n_{CCN}$ and 3 for $n_{INP}$ (not indicated). The immersion freezing INP parameterization of D15d for dust at a constant temperature is used to give an estimate. CCN concentrations are given for 0.2% water supersaturation (ss). $n_{CCN}$ and $n_{INP}$ values for the observed dust–smoke mixture and the pure marine conditions (INP from M18m) measured at Barbados on 3 March 2014 and 26 February 2014, respectively, are added.

| Date | Height | $n_{CCN}$ **Falcon** 0.2% ss | $n_{CCN}$ **Lidar** 0.2% ss | $n_{INP}$ **Lidar** D15d –20°C | $n_{INP}$ **Lidar** D15d –25°C | $n_{INP}$ **Lidar** D15d –30°C |
|---|---|---|---|---|---|---|
| | km | cm$^{-3}$ | cm$^{-3}$ | L$^{-1}$ | L$^{-1}$ | L$^{-1}$ |
| 22 June 2013 | 2 – 3 | 158±13 | 242±74 | 7±3 | 74±30 | 753±303 |
| | 3 – 3.6 | 88±6 | 144±21 | 3±1 | 25±5 | 259±49 |
| 10 July 2013 | 2 – 3 | 157±13 | 291±12 | 19±2 | 196±20 | 1993±206 |
| | 3 – 4.4 | 100±5 | 189±22 | 9±1 | 88±12 | 896±128 |
| 11 July 2013 | 2 – 3 | 154±11 | 270±21 | 15±1 | 149±13 | 1488±126 |
| | 3 – 4.4 | 107±7 | 196±18 | 9±1 | 92±12 | 918±120 |
| 26 February 2014 | 0.5 – 1.5 | – | 166±67 | 0.004±0.002* | 0.06±0.03* | 0.9±0.4* |
| 3 March 2014 | 2 – 3 | – | 412±62 | 3±1 | 32±11 | 330±112 |

* INP concentration calculated with McCluskey et al. (2018) for marine particles

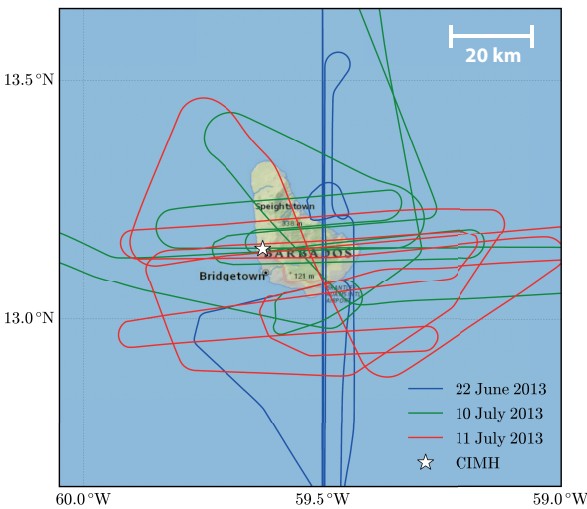

**Figure 1.** Falcon flight tracks in the Barbados region on 22 June, 10 July, and 11 July 2013. The white star marks the lidar site at the Caribbean Institute for Meteorology and Hydrology (CIMH) north of the capital Bridgetown. Falcon aircraft versus BERTHA lidar comparisons are based on the observations listed in Table 4.

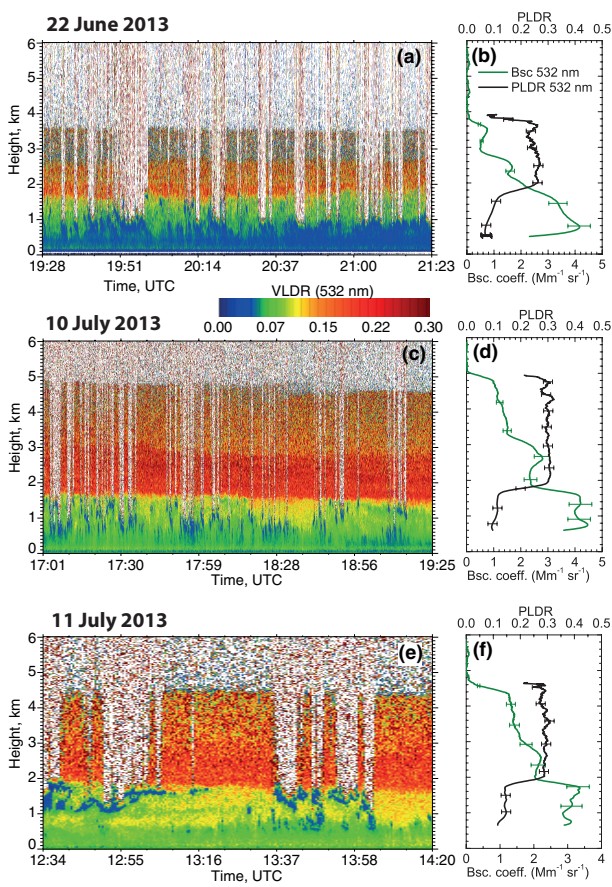

**Figure 2.** SALTRACE lidar observations of the Saharan air layer (SAL) above the marine boundary layer on 22 June (a-b), 10 July (c-d), and 11 July (e-f) 2013. Time-height displays of the volume depolarization ratio at 532 nm (left) and the corresponding cloud-screened mean profiles (right) of the particle backscatter coefficient (green line, lower x-axis) and particle linear depolarization ratio (black line, upper x-axis) at 532 nm are shown. Low-level trade wind cumuli (dark blue in a, c, e) strongly attenuated the laser light, indicated by the noise above the clouds. The strong increase of the depolarization ratio indicates the lower boundary of the SAL at approx. 1.8–2.0 km height. The top of the SAL was about 3.7 km (22 June), 5.0 km (10 July), and 4.5 km (11 July). Local time is UTC –4 h.

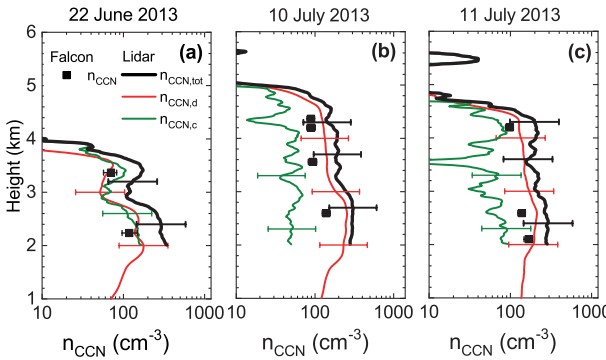

**Figure 3.** Lidar-derived CCN number concentrations at 0.2% supersaturation (black line) with contributions from dust (red line, critical dry diameter of 200 nm) and continental pollution aerosol (olive line, critical dry diameter of 100 nm) compared to coincident airborne in situ measurements (black dots) during SALTRACE-1. The error bars of the lidar profiles indicate an uncertainty of a factor of 2. The error bars of the in situ measurements indicate the 16[th] and 84[th] percentile.

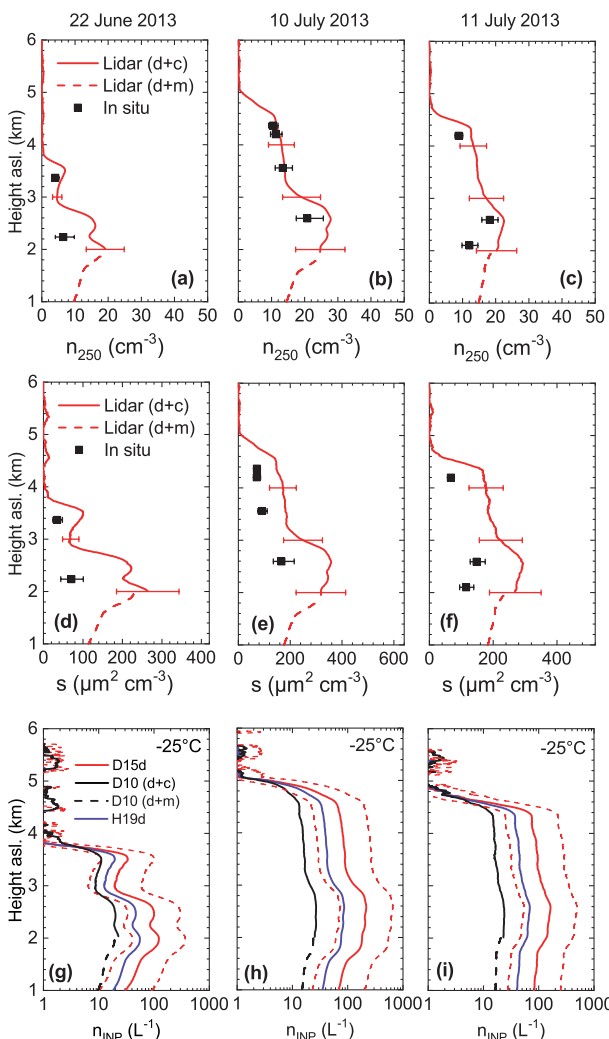

**Figure 4.** Number concentration $n_{250}$ for particles with radius > 250 nm (a–c) and surface area concentration $s$ (d–f) measured on board the Falcon aircraft (black dots) and derived from the lidar measurements (red profiles, solid line – sum of dust and continental pollution particles (above 2 km), dashed line – sum of dust and marine particles (below 2 km)). The three SALTRACE case studies are shown: 22 June (left, a,d,g), 10 July (center, b,e,h), and 11 July 2013 (right, c,f,i). INP concentrations (g–i) are given at −25°C for the immersion freezing parameterizations of D10d+c (input $n_{250,d}+n_{250,c}$, above approx. 2 km), D10d+m (input $n_{250,d}+n_{250,m}$, below approx. 2 km), D15d (input $n_{250,d}$), and H19d (input $s_d$, see Tab. 2). 1% K-feldspar contribution was used for H19d. The uncertainty in the lidar-derived $n_{250}$ and $s$ values is 30%. For the INP concentration an uncertainty of a factor 3 is indicated by the dashed lines for the D15d profile. The error bars of the in situ measurements indicate the 16[th] and 84[th] percentile.

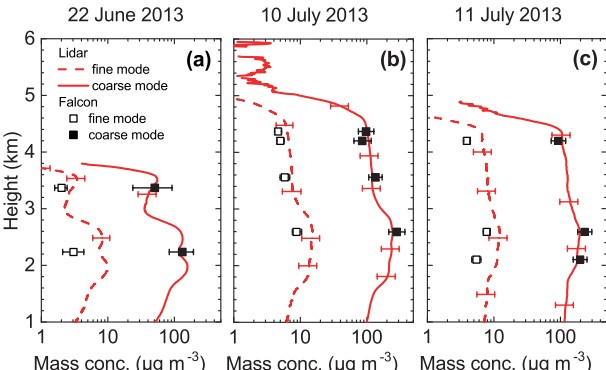

**Figure 5.** Mass concentration of fine mode ($r$<500 nm, dashed line) and coarse mode ($r$>500 nm, solid line) dust derived from airborne in situ measurements (black dots) and lidar observations (red profiles) for the three SALTRACE-1 days. The error bars of the in situ measurements indicate the 16$^{th}$ and 84$^{th}$ percentile.

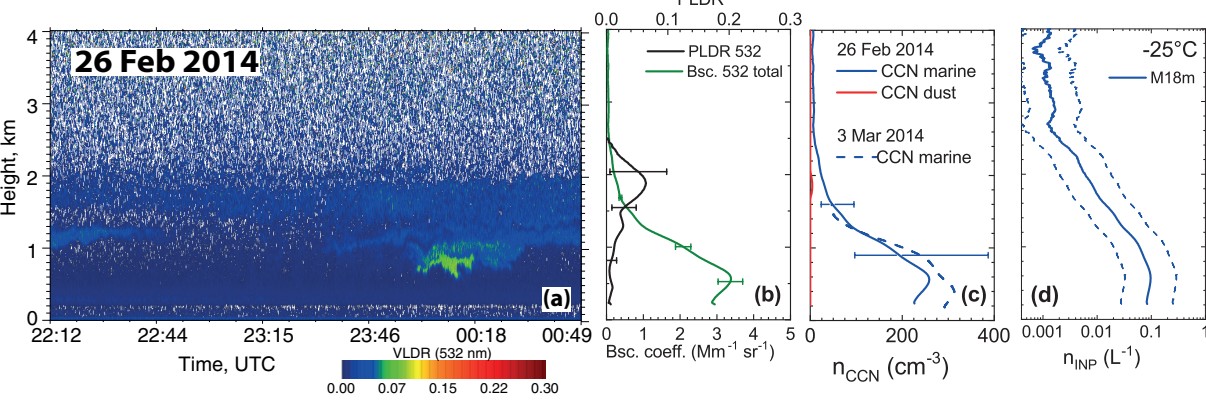

**Figure 6.** Pristine marine observation during the SALTRACE winter campaign on 26 February 2014, 22:12–00:49 UTC. (a) Time-height display of the 532 nm volume depolarization ratio, (b) particle backscatter coefficient (green line) and particle linear depolarization ratio (black line) at 532 nm, (c) CCN number concentrations at water supersaturation of 0.2%, (d) immersion freezing INP concentrations at –25°C for M18-marine. The dashed line indicates the uncertainty range (factor 3) of the INP parameterization.

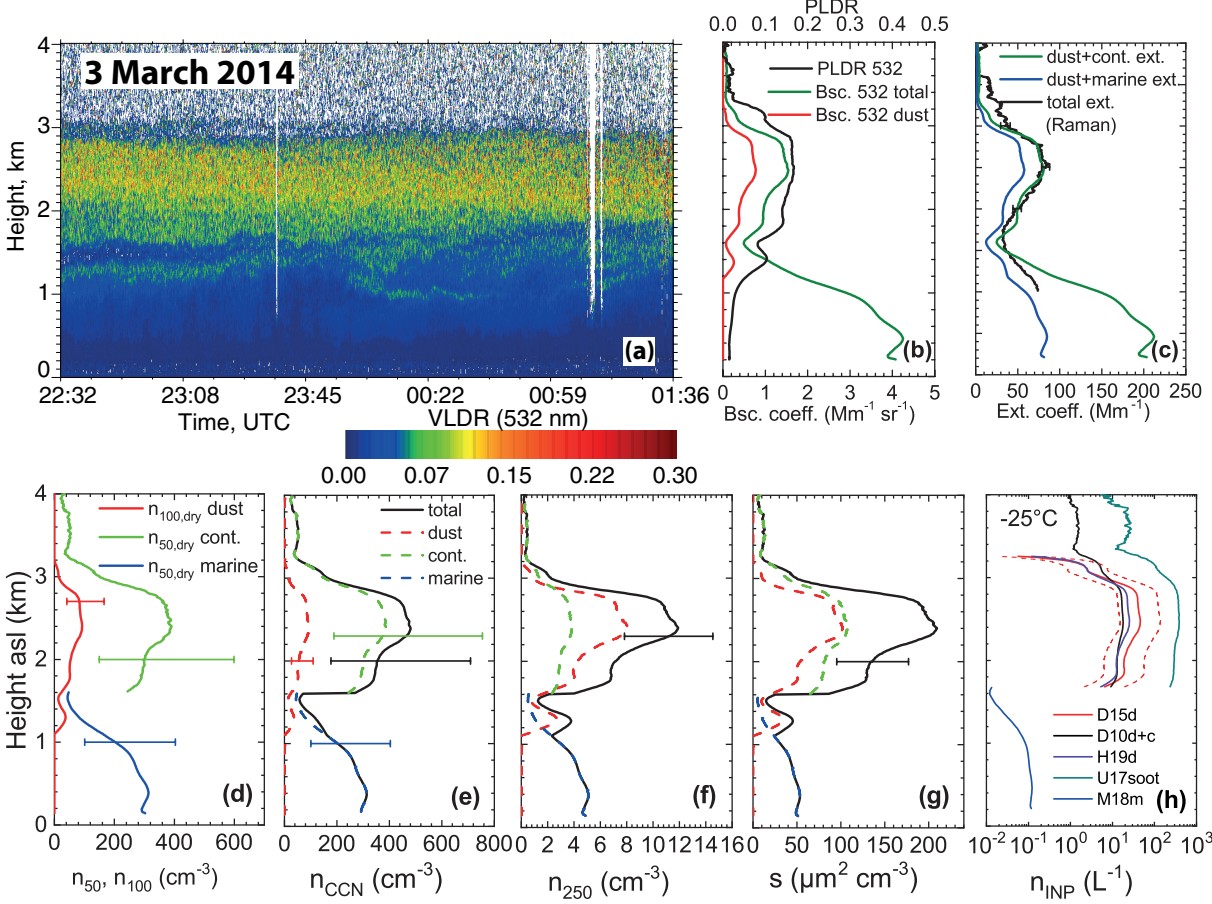

**Figure 7.** Dust–smoke mixture observed during the SALTRACE winter campaign on 3 March 2014, 22:30–23:20 UTC. (a) Time-height display of the 532 nm volume linear depolarization ratio (VLDR) (only the first 40 min are averaged for the profiles in b–h), (b) particle backscatter coefficient (green line) including its dust contribution (red line) and particle linear depolarization ratio (black line) at 532 nm, (c) sum of dust and continental pollution extinction coefficient (green line) using a smoke lidar ratio of 50 sr and sum of dust and marine particles extinction coefficient (blue line) using a marine lidar ratio of 20 sr compared to the total extinction coefficient (black line) independently measured with BERTHA (Raman lidar method). Above the height of 1.6 km a dust–smoke mixture fits best, below the dust–marine mixture (with a small contribution of smoke or pollution) agrees better with the Raman extinction solution. (d) Number concentration $n_{100,d}$ for dust (red), $n_{50,c}$ for smoke (green), $n_{50,m}$ for marine particles (blue), (e) CCN number concentrations at water supersaturation of 0.2% for the 3 components and the total CCN concentration (black line) above 1.6 km for dust–smoke, below for dust–marine, (f) $n_{250}$ values (colors as before), (g) surface area concentration (colors as before), (h) immersion freezing INP concentrations at –25°C for D10-cont+dust, D15-dust, H19-dust, M18-marine and U17-soot. For the INP concentration an uncertainty of a factor 3 is indicated as dashed lines for the D15d profile.

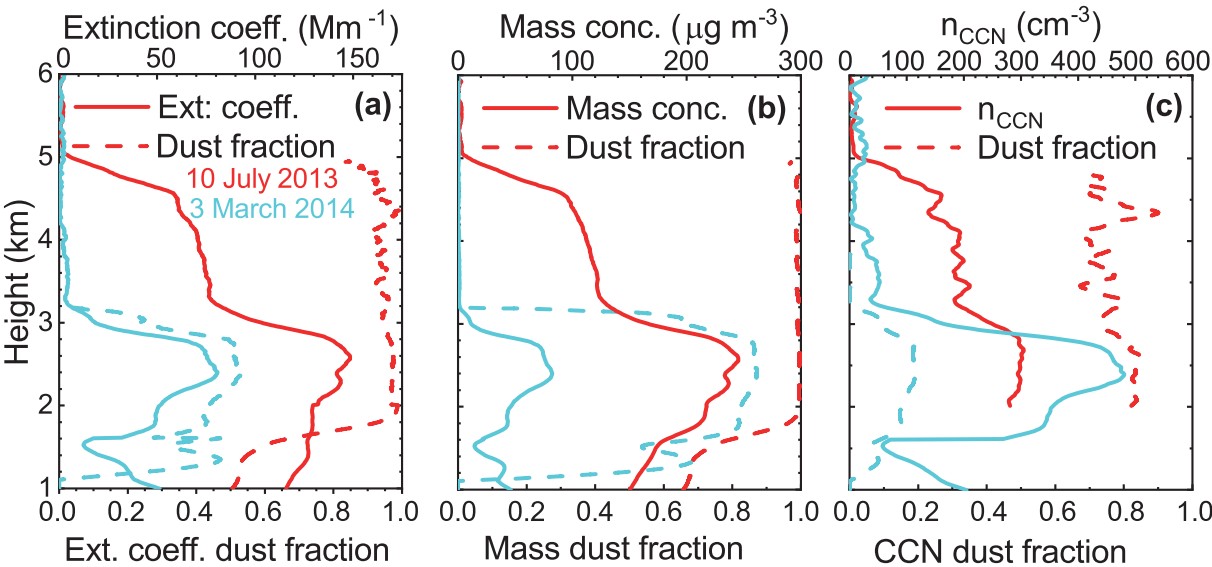

**Figure 8.** Summer (10 July 2013, red) versus winter (3 March 2014, cyan) aerosol conditions in the SAL. (a) total particle extinction coefficient (solid line) and relative dust contribution to the total particle extinction coefficient (dashed line), (b) same as (a) except for dust mass concentration, (c) same as (a) except for CCN concentration.