# Peer review of "Profiles of cloud condensation nuclei, dust mass concentration, and INP-relevant aerosol properties in the Saharan Air Layer over Barbados from polarization lidar and airborne in situ measurements"

_Atmospheric Chemistry and Physics, 2019_

## Referee Comment (RC1) · Paul DeMott (Referee) · 19 Jun 2019

**General Comments**

This paper continues a series of papers themed around the use of lidar retrievals to estimate CCN and INP profiles. These developments of capabilities are being followed with interest by a broad community since the applications and utility are obvious for regions where in situ data are not available, or not available with high frequency. This paper will make a nice incremental contribution to the growing literature base of this team, focusing here on comparison to aircraft data that did not have INP data to compare to. I do have some critical comments and suggestions in a few regards.

[Figure]

1) First, I believe that the nature of retrieval of multiple species contributions to aerosol number, mass, CCN and INP requires some additional description because this is a more recent development (versus retrievals from layers dominated by a single aerosol type) and so bears reiterating from its introduction over the last two years. If the Marinou et al. paper is accepted for publication, reference should be made to the detailing of the detailed schematic there.

2) Secondly, I feel that the use of the DeMott et al. (2010) parameterization as specific to continental and non-dust contributions to INPs is not exactly correct, and this has implications. The "continental" definition in Mamouri and Ansmann (2016) neglects the fact that dust contributions to INPs were most certainly folded into the parameterization in a variety of environments. I see now that Marinou et al. (2018) have written, "As the majority of the samples used for D10 are non-desert continental aerosols, this INP parameterization has been considered to be suitable for addressing the immersion and condensation freezing activity of mixtures of anthropogenic haze, biomass burning smoke, biological particles, soil and road dust (Mamouri and Ansmann, 2016)." This is also a gross simplification, with the actual contributors unknown, and the likelihood that dust was folded in at a variety of levels of contribution. After all, one study was PACDEX, the Pacific Dust Experiment. Hence, strong caveats about potential duplication of INPs, and lack of assured attribution to all of the other types mentioned, are needed here. What one may really wish for are parameterizations for all relevant INP species instead. Substitution of D10 for the absence of such detailed information is not ideal, and so I am concerned that this is being glossed over. It is worrisome that this assumption seems to have propagated into a number of papers since 2016, and in some cases is even called "non-dust" or continental "pollution", the former not being true to the original paper and the latter being a true stretch in attribution that has never been supported by direct evidence.

3) I also wondered about the use of the groups' own parameterization of sea spray aerosols (based on DeMott et al., 2016, since that paper did not promote a direct

parameterization) versus a marine-specific parameterization for the Atlantic region that is referenced in the introduction (McCluskey et al., 2018). Do they compare well? I obviously know the answer, but you might justify persisting with a parameterization that did not as deeply consider "pure" marine as did the newer McCluskey paper. I realize that this is a very minor point, since marine INPs at -25 °C are minor contributors compared to mineral dusts in SAL conditions.

4) For the use of the DeMott et al. (2015) parameterization (D15), it seems that a decision has been made to not use the recommended 3x correction factor for immersion freezing that was justified in that paper? If so, the basis/reasoning for this should be stated.

5) Finally, I think that it would be very useful to demonstrate retrievals in a profile that does not necessarily include dust or smoke overlying or mixing in the region above the marine boundary layer. That would represent the unperturbed case, and give insights into the behavior of the combined set of parameterizations when dust is not at all dominant.

Additional context to these comments and some additional specific questions/editorial comments for addressing before publication are listed below.

**Specific Comments**

1) Page 2, lines 2-4: What papers are you referring to in stating the implementation of these parameterization schemes? These are not all included in this present paper, although it would be interesting to see. Also, please note that there is no parameterization given in DeMott et al. (2016). This must have been created by the authors.

2) Page 2, line 10: fix "several 10000 km" to state a range of distances expected.

3) Page 2, lines 25-27: Note that as written, the sentence is repetitive in mentioning dust and smoke mixture at the beginning and ends of the sentence.

4) Page 3, lines 2: The continental aerosol designation is not mentioned here, as listed

in Table 2. As stated above, this needs some serious caveats applied, namely that it is used in the absence of a true set of parameterizations that could describe other than mineral dust input, even though it definitely includes some influence from varied levels of mineral dusts in the studies used by D10. It was not intended to be specific or neglectful of any particular class of INPs.

5) Last paragraph of Page 3, and start of Page 4: This discussion of assumptions on the hygroscopicity of mineral dust wanders some and never quite makes clear if kappa values for Saharan dust after transport to the region have been measured as low as is assumed or if this is an assumption based on the "fresh" nature of dust observed via say, microscopy studies. There is a difference, as trace amounts of materials can make a difference. In the end, it seems that the value selected of 0.02 is in the range of most measurements (i.e., not fully hydrophobic), and in the range estimated to be consistent with activation in clouds as submicron dust particles in the Eastern Atlantic (Twohy et al., GRL, 36, L01807, doi:10.1029/2008GL035846, 2009).

6) Page 4, line 14: The statement "The very hydrophilic sea salt particles (sodium chloride) have an activation diameter..." sounds awkward. Sea salt is hygroscopic. But sea salt is rarely the composition of sea spray particles alone, so why not say that "We assume a composition of sea salt for marine aerosols, and prescribe an activation diameter of..."

7) Page 5, line 23: Perhaps discuss that cumuli attenuate the lidar, versus "disturbed" the measurements?

8) Page 5, paragraph starting line 29: This is where I suggest that some elaboration on the methods for retrieving the contributions of different aerosols in a mixed scenario is given.

9) Page 6, line 21: the CCN data from the Falcon are "measurements." They may have uncertainties, but they are not retrievals.

10) Page 7, first paragraph: Is the surface area used only for the marine parameterization? Do the dust parameterizations using s significantly differ from D15? I only wondered about the derivation of surface area if it was not going to be used.

11) Page 7, lines 8-10: This is a rather subjective statement about the likely role of the INP concentrations derived for the SAL. Clearly, direct cloud observations or cloud model simulations are likely needed to explore the implications, since such tropical cumuli are known to contain rather vigorous secondary ice formation processes through their deep supercooled layers (e.g., see Lasher-Trapp et al., J. Atmos. Sci., 73, 2547-2564, 2016, and references to Lawson et al., 2015 and Heymsfield and Willis, 2014 therein).

12) Page 7, line 21: Consider replacing "Wrong in situ particle counting. . ." with uncertainties in in situ aerosol measurements. There is no support provided for how or why the measurements would be wrong. They are your only link to apparent ground truth.

13) Page 8, line 15: suggest leading to "likely changes in trade wind cumulus cloud microphysical properties. . ." rather than "developments". Also, does it not depend on which layer dominates aerosol contributions to convective clouds?

14) Page 8, line 32: Suggest "reconciled" for "fixed"

15) Page 9: The summary paragraph is a bit short in its outlook for the future. You would seem to benefit from more validation INP data, particularly for cases with and without dust, so the validity of the apparent knowledge of continental that you promote is also checked. And not only in dusty situations. Will you have INP data in any of the forthcoming campaigns? Quantifying other specific aerosol type contributions than dust and marine would appear useful as well.

16) Table 1 header: the cv coefficients need explanation. Are these the "conversion factors" mentioned?

17) Table 2: as mentioned, DeMott et al. (2016) does not include a parameterization, so

that is not the appropriate reference for it. I suggest using the specific parameterization of McCluskey et al., if that is possible. Also the D15d reference needs to mention somewhere (if not in the table) what cf factor is used in this study.

18) Figure 4: I can note similar s values here as in DeMott et al. (2015) for the SAL over the Western Caribbean, but the predicted INP concentrations are a bit lower here at around -25°C. This motivated me to ask about the cf factor assumed for use in the parameterization in the present study. Also, below 2km, is it certain you are dealing with dust and not marine aerosols in all cases? Is this why the lidar profile showing higher surface area and n250 on 22 June still leads to a decrease in D15-predicted INPs? Is that because you presume all of those particles are "continental"? This is where I think the application of a D10+D15 approach could lead to errors, and the only way to tell will be future in situ INP measurements.

19) Figure 6 inspired me to ask what an unperturbed profile might look like, for example when there is not a strong dust or smoke or pollution layer over the clouds. Do you have any such data?

---

## Referee Comment (RC2) · Anonymous Referee #2 · 24 Jun 2019

**Review of "CCN concentration and INP-relevant aerosol profiles in the Saharan Air Layer over Barbados from polarization lidar and airborne in situ measurements" by M. Haarig et al.**

**General Comments**

This paper describes the use of lidar measurements to retrieve profiles of CCN and IN in the Saharan Air Layer over Barbados. It continues the series of paper published by this group that describe the use of these lidar measurements to retrieve particle properties using lidar measurements and to use these measurements to estimate CCN and IN. The paper compares the lidar retrievals with airborne in situ measurements provides to give some indication of how well these lidar retreivals work. Although the lidar retrieval of CCN has large uncertainty, it still provides some utility for estimating the vertical distribution of CCN. I suggest publication after the authors address the minor comments below.

On page 3 and elsewhere, the authors mention the use of appropriate extinction-to-backscatter values for dust, marine aerosol, and continental aerosol. It looks like these specific values are then used to convert backscatter to extinction for use in the retrieval algorithms. However, it's not clear why these specific values are used when the Raman lidar measurements actually provide the means to directly measure aerosol extinction (as well as the extinction-to-backscatter ratio). Why not use the actual Raman lidar measurements rather than these specific values? Is the SNR too low to directly retrieve these parameters during the daytime?

**Specific Comments**

1. Abstract, line 5. Suggest "….properties measure in situ with aircraft…"
2. Abstract, lines 7-8, What is meant by "reasonable agreement" between lidar and in situ number concentrations? More quantitative description would be useful.
3. Page 5, line 29. Does the method used by Ansmann et al. (2017) to decide whether the non-dust component was marine or continental use actual measurements of the lidar ratio? It would be helpful to have additional information here.
4. Page 6, lines 11-12. How was the lidar retrieval uncertainty determined to be a factor from 2 to 3? Where did this come from?
5. Page 6, line 13. "Besides the large retrieval uncertainty, other uncertainty sources may have contributed to the systematic bias between the lidar and airborne in situ observations:" The following sentences then describe other uncertainty sources. It's not clear what sources of error contribute to the factor of 2-3 lidar uncertainty which are separate from the other uncertainty sources described in the following sentences. How much additional uncertainty do these other sources add to the factor of 2-3 lidar retrieval uncertainty?
6. Page 7, line 20. What is meant by "large disagreement"
7. Table 1. Near the bottom, the formula for $n_{CCN}$ contains items $f_{ss,d}$, $f_{ss,c}$, and $f_{ss,m}$. How are these factors determined?
8. Figure 1. Suggest replacing the flight numbers in the legend with the dates of the three flights.
9. Figure 2. The color images show range-corrected signals of the cross-polarized channel. Why not instead show images of the actual particulate lilnear depolarization ratio? This would make it easier to compare the results from day to day.
10. Figure 6. Same comment as item 9 above.

---

## Referee Comment (RC3) · Anonymous Referee #3 · 16 Jul 2019

The manuscript presents case study results from a few flights of the SALTRACE campaign, where ground-based Raman lidar measurements were made coincident with airborne in situ aerosol measurements. The Raman lidar backscatter measurements are converted to extinction coefficients by using an assumed lidar ratio, and then these extinction coefficients are used to estimate particle number concentrations using empirical conversion factors from previous literature. It appears from the manuscript that only three different aerosol types are considered, which are largely distinguished by whether the aerosols are depolarizing (indicating dust) or not depolarizing (indicating continental pollution). It's not clear how marine particles are identified, as these particles are likely to depolarizing when dry, but non-depolarizaing when hydrated.

Having estimated particle number concentrations, then CCN and INP concentrations for arbitrary cloud conditions (e.g., 0.2% supersaturation) are estimated using additional assumed CCN=f(s) and INP activation functional relationships. All in all, there are a lot of assumptions made to get from Point A (lidar backscatter) to Point B (CCN and INP concentrations) and quite a lot of uncertainties stacking on top of each other. While the mass profile comparisons look great (Fig. 5), the agreement among the number concentration comparisons is much less strong. These relationships have been seen before in prior literature that have used more rigorous retrieval algorithms that rely on many fewer empirical assumptions (e.g., Sawamura et al., ACP, 2017, https://www.atmos-chem-phys.net/17/7229/2017/). While the present paper examines data from a few cases of merit, the methods represent only a incremental science contribution that doesn't really seem to advance the state-of-the-art. I defer to the editor's judgment as to whether this is sufficient to merit publication in ACP.

Specific comments:

1) The statement on Pg. 2, Line 9 that the ground-based lidar is observing CCN number concentration and INP-relevant aerosol properties is not true! No such measurement is being made. Instead, a highly-empirical series of conversion factors are being applied to backscatter observations to retrieve aerosol concentrations that may or may not be relevant for CCN and INP activation processes. In addition, it needs to be recognized that there is already quite a bit of literature looking at relationships between lidar measurements and aerosol extensive parameters. It doesn't seem appropriate to imply that this study is somehow "a first", as to make the case for this, one has to slice the data attributes pretty finely (e.g., use of a specific ground-based Raman lidar and the focus on dust over the remote Atlantic west of the source regions, conversion of particle number concentration measurements to CCN and INP concentrations at specific, arbitrary conditions). There needs to be better truth in advertising in this paper. Please remove the sentences on Pg. 2, Line 9; Pg. 8, Line 29; and potentially elsewhere that imply that this study is a first of its kind and that the lidar is observing CCN and INPs.

2) What is the basis for doing the CCN comparisons at 0.2% supersaturation? Is this a realistic supersaturation for clouds in this region? How would the comparison look for higher supersaturations (e.g., 0.4% or 0.6%)? For lower supersaturations (e.g., 0.1%)?

3) After reading the conclusions section, I'm unclear how this study advances the use of lidar observations to place constraints on CCN or INPs. The outlook that the authors lay out is that more comparisons in other environments are needed. Why? How will more comparisons be helpful? Would we expect the agreement betweent the lidar retrieval and in situ data to be better or the same? What contribution does this study make? I'd like to see more discussion that contextualizes how the present study is an advance upon the state-of-the-art.
* * *

---

## Author Comment (AC1) · 25 Sep 2019

**Author's final response to the reviewer comments**

We want to thank the two anonymous reviewers and Paul DeMott for their time and effort spend to improve the manuscript. Their helpful comments are incorporated into the paper and will be addressed in more detail below. The main changes mostly motivated by the comment of the reviewers in the manuscript are:

- The title has been changed to: "Profiles of cloud condensation nuclei, dust mass concentration, and INP-relevant aerosol properties in the Saharan Air Layer over Barbados from polarization lidar and airborne in situ measurements"
- The introduction of the McCluskey et al. (2018) parameterization for marine aerosol.
- The addition of the recently published Harrison et al. (2019) parameterization for dust particles (K-feldspar with 1% contribution to the dust surface area).
- The use of the DeMott et al. (2010) was modified to use it for all aerosol (and not as a non-dust parameterization).
- A new subsection 5.1 was added to present a pure marine case from the SALTRACE-2 winter campaign to show unperturbed conditions.

The referee's comments are shown in *italic* and our responses are added in **bold**.
The manuscript with our changes marked in bold is attached.

**Reply to Anonymous Referee 2**

*General Comments*
*This paper describes the use of lidar measurements to retrieve profiles of CCN and IN in the Saharan Air Layer over Barbados. It continues the series of paper published by this group that describe the use of these lidar measurements to retrieve particle properties using lidar measurements and to use these measurements to estimate CCN and IN. The paper compares the lidar retrievals with airborne in situ measurements provides to give some indication of how well these lidar retrievals work. Although the lidar retrieval of CCN has large uncertainty, it still provides some utility for estimating the vertical distribution of CCN. I suggest publication after the authors address the minor comments below.*

*On page 3 and elsewhere, the authors mention the use of appropriate extinction-to-backscatter values for dust, marine aerosol, and continental aerosol. It looks like these specific values are then used to convert backscatter to extinction for use in the retrieval algorithms. However, it's not clear why these specific values are used when the Raman lidar measurements actually provide the means to directly measure aerosol extinction (as well as the extinction-to-backscatter ratio). Why not use the actual Raman lidar measurements rather than these specific values? Is the SNR too low to directly retrieve these parameters during the daytime?*
**Let us explain the used process in more detail:**
**The separation of the different aerosol types is done for the particle backscatter coefficient using the particle depolarization ratio measurement.  Then, we have to convert the backscatter coefficient to an extinction coefficient using the lidar ratio (based on literature, now cited in the manuscript). The total particle extinction obtained by this process can be verified by the extinction obtained independently by the Raman method. For daytime measurements, we take the closest nighttime measurement and test the aerosol type separation procedure.**
**In conclusion, the main reason for not using directly the extinction coefficient from the Raman method is the aerosol type separation process. A aerosol type separation process directly for the extinction coefficient may be developed in future.**
**We updated the manuscript in Section 2.1 and 3 to make this point better comprehensible.**

*Specific Comments*
*1. Abstract, line 5. Suggest "….properties measure in situ with aircraft…"*
**Changed.**

*2. Abstract, lines 7-8, What is meant by "reasonable agreement" between lidar and in situ number concentrations? More quantitative description would be useful.*
**The sentence was reformulated as follows:**
**The CCN number concentrations derived from lidar observations were up to a factor of two higher than the ones measured in situ on board the research aircraft Falcon. Possible reasons for the difference are discussed.**

*3. Page 5, line 29. Does the method used by Ansmann et al. (2017) to decide whether the non-dust component was marine or continental use actual measurements of the lidar ratio? It would be helpful to have additional information here.*
**Thank you for the comment. We see that we have to describe this procedure in a better way.**
**The described method makes use of the independently measured extinction coefficient with the Raman lidar method (nighttime only). The extinction coefficient calculated from the dust backscatter coefficient (multiplied by the literature-based dust lidar ratio) and the non-dust backscatter coefficient (either multiplied by the typical marine (20 sr) or continental (50 sr) lidar ratio) has to be the same as the independently measured Raman (total) particle extinction coefficient.**
**The corresponding part was newly formulated to better explain the method.**

*4. Page 6, lines 11-12. How was the lidar retrieval uncertainty determined to be a factor from 2 to 3? Where did this come from?*

**Thank you for pointing out this issue. An explaining sentence was added.**

**"The retrieval uncertainty results from the uncertainty in determining the extinction-to-number-concentration conversion factor for small particles (r>50 nm or r>100 nm) using AERONET derived AOD and columnar number concentrations (n50;n100) as described in Mamouri and Ansmann (2016)."**

**It is the uncertainty of the fit in the logarithmic space as shown in Mamouri and Ansmann (2016), Fig. 4,5, and 6. For the larger particles (n250) and the surface area concentration, the regression can be done in the linear space resulting in lower uncertainties.**

*5. Page 6, line 13. "Besides the large retrieval uncertainty, other uncertainty sources may have contributed to the systematic bias between the lidar and airborne in situ observations:" The following sentences then describe other uncertainty sources. It's not clear what sources of error contribute to the factor of 2-3 lidar uncertainty which are separate from the other uncertainty sources described in the following sentences. How much additional uncertainty do these other sources add to the factor of 2-3 lidar retrieval uncertainty?*

**The factor 2 is the retrieval uncertainty resulting from the AERONET-based conversion factors (Mamouri and Ansmann, 2016) covering the basic question: How many n100 or n50 particles do we expect for a given extinction coefficient and a given particle type? Further sources of errors for our specific measurements are then listed in the paper.**

**Combined with the answer to your comment number 4, this should be stated clearer now in the manuscript.**

*6. Page 7, line 20. What is meant by "large disagreement"*

**The text was reformulated to avoid unclear formulations:**

**"Again for the fine particle dominated quantities, i.e., the fine-mode mass concentration, the lidar derives higher values than the in situ observed ones."**

*7. Table 1. Near the bottom, the formula for nCCN contains items fss,d, fss,c, and fss,m. How are these factors determined?*

**An explanation have been added in the text (Sect. 2.1, page 4):**

**"An enhancement factor fss determined in Mamouri and Ansmann (2016) from various laboratory and field studies (activation diameter and supersaturation) is used to retrieve nCCN for different supersaturation levels (see Table 1)."**

*8. Figure 1. Suggest replacing the flight numbers in the legend with the dates of the three flights.*

**Replaced.**

*9. Figure 2. The color images show range-corrected signals of the cross-polarized channel. Why not instead show images of the actual particulate linear depolarization ratio? This would make it easier to compare the results from day to day.*

**Thank you for the suggestion. The plots (in Fig. 2,6,7) now show the volume linear depolarization ratio, where the dust layer is much better visible than in the range-corrected signal.**

*10. Figure 6. Same comment as item 9 above.*
**Done.**

**Reply to Anonymous Referee 3**

*The manuscript presents case study results from a few flights of the SALTRACE campaign, where ground-based Raman lidar measurements were made coincident with airborne in situ aerosol measurements. The Raman lidar backscatter measurements are converted to extinction coefficients by using an assumed lidar ratio, and then these extinction coefficients are used to estimate particle number concentrations using empirical conversion factors from previous literature. It appears from the manuscript that only three different aerosol types are considered, which are largely distinguished by whether the aerosols are depolarizing (indicating dust) or not depolarizing (indicating continental pollution). It's not clear how marine particles are identified, as these particles are likely to depolarizing when dry, but non-depolarizing when hydrated.*

**Let me give some explanations to the legitimate concerns of the reviewer. The used lidar ratios are confirmed by several studies based on Raman lidar measurements, e.g., Mueller et al., 2007, Groß et al., 2013, Baars et al., 2016 (now cited in the manuscript). And independent Raman extinction measurements are used to check the contributions of the non-dust aerosol (marine or continental). This process is now described in more detail at the end of Section 3, and was introduced and illustrated in Ansmann et al. (ACP, 2017). The key point here is that marine particles cause a lidar ratio of about 20 sr and fine-mode dominated particles (urban haze, biomass burning smoke) cause about 50 sr at 532 nm. And that helps to decide whether the non-dust component is of marine or continental origin as shown by Ansmann et al. (2017).**

**A slight influence of dry marine particles could be present, but the effect should be negligible inside the SAL as the air masses are almost undisturbed since leaving the African continent. The newly added case of pure marine conditions (Section 5.1), where no aerosol transport from Africa was present, shows a layer with some dry marine particles on top (Fig. 6), which can indeed lead to a misclassification. However, the effect on CCN is very small.**

*Having estimated particle number concentrations, then CCN and INP concentrations for arbitrary cloud conditions (e.g., 0.2% supersaturation) are estimated using additional assumed CCN=f(s) and INP activation functional relationships. All in all, there are a lot of assumptions made to get from Point A (lidar backscatter) to Point B (CCN and INP concentrations) and quite a lot of uncertainties stacking on top of each other. While the mass profile comparisons look great (Fig. 5), the agreement among the number concentration comparisons is much less strong. These relationships have been seen before in prior literature that have used more rigorous retrieval algorithms that rely on many fewer empirical assumptions (e.g., Sawamura et al., ACP, 2017, https://www.atmos-chem-phys.net/17/7229/2017/). While the present paper examines data from a few cases of merit, the methods represent only an incremental science contribution that doesn't really seem to advance the state-of-the-art. I defer to the editor's judgment as to whether this is sufficient to merit publication in ACP.*

**Indeed a lot of empirical assumptions are necessary, but they are clearly justified and based on climatological facts (AERONET observations and the correlation between particle extinction coefficient and particle number concentration). Our method is straight forward and the scatter in the AERONET plots clearly indicates the uncertainty. We have a well-defined error range for each of our parameters. So what is the point of criticism? In contrast, these rigorous retrieval algorithm, the reviewer mentions, are based on multiwavelength lidar methods and these methods belong to the class of ill-posed problems, and a lot of constraints need to be introduced to stabilize the solutions. Ill posed means even… almost undefined uncertainty range. That was the main reason to search for alternatives. Our method is much more robust than the multiwavelength method, especially when taking into account what immense effort is needed to guarantee always high quality multiwavelength lidar measurements over days, months and years. This is simply not possible. There is a mix of photoncounting and analog detection channels. This already often kills homogeneity and consistency in the data sets needed in these inversion methods, mentioned by the reviewer. In addition, there is notoriously the problem with 1064 nm: How to calibrate? How accurate are the 1064 nm backscatter coefficients? This is always an open question, even in the case of CALIOP. So, the multiwavelength lidar is often not usable as our experience with EARLINET over the last 20years clearly indicates. We need alternatives, methods that are robust, simple, straight forward and applicable to simple**

and robust lidar measurements. And exactly that was the basic motivation why we came up with polarization lidar method used in this study. This is an exciting and simple approach and it works!

So in conclusion, we prefer our method, but sure…. to check the quality of the products we need independent in situ measurements. Therefore we need aircraft observations. They are necessary to check to what extent all the conversions work or not. And our Barbados observations and comparisons are of high value because here we have aged dust after long-range transport over more than 5000 km. It is not easy to find other locations to measure such undisturbed scenarios as presented in this paper.

Thank you for pointing to the work of Sawamura et al. (2017). One could argue our work is somehow complementary to their study as they had to exclude all periods influenced by dust or other non-spherical particles, and we are focusing exactly on dust in the Saharan Air Layer. Lidar inversions of spherical particles have been widely applied, but non-spherical dust particles remain challenging for the retrievals. The AERONET-based approach of the POLIPHON method (Mamouri and Ansmann, 2016) overcomes the difficulties of the non-sphericity problem in the case of dust particles. We do not need no questionable dust shape model (spheroidal shape model) as needed in respective multiwavelength inversion attempts.

Our observations are unique and clearly deserve publication. To do field campaigns in the Caribbean with aircraft and advanced ground-based aerosol lidar is not just a routine job. SALTRACE stands not only for dust investigation in the Saharan air layer (SAL), but also for aerosol-cloud interaction experiment (…ACE). And this study here is clearly a contribution to this field of research. There is no publication with the topic we focus on. We focus on CCN and INP-relevant aerosol parameters. This is new! At least, we have never seen any publication except the TROPOS/CUT papers dealing with INP profiles.

The three presented cases have been selected based on the vicinity of Falcon aircraft observations to the lidar site and more cases would be desirable. Nevertheless, the similar good agreement of the three presented cases makes us believe that more comparisons in the SAL around Barbados would lead to the same results. The careful analyzed and quality assured data sets from the in situ and lidar measurements are a solid basis for the inter comparison study.

*Specific comments:*

*1) The statement on Pg. 2, Line 9 that the ground-based lidar is observing CCN number concentration and INP-relevant aerosol properties is not true! No such measurement is being made. Instead, a highly-empirical series of conversion factors are being applied to backscatter observations to retrieve aerosol concentrations that may or may not be relevant for CCN and INP activation processes. In addition, it needs to be recognized that there is already quite a bit of literature looking at relationships between lidar measurements and aerosol extensive parameters. It doesn't seem appropriate to imply that this study is somehow "a first", as to make the case for this, one has to slice the data attributes pretty finely (e.g., use of a specific ground-based Raman lidar and the focus on dust over the remote Atlantic west of the source regions, conversion of particle number concentration measurements to CCN and INP concentrations at specific, arbitrary conditions). There needs to be better truth in advertising in this paper. Please remove the sentences on Pg. 2, Line 9; Pg. 8, Line 29; and potentially elsewhere that imply that this study is a first of its kind and that the lidar is observing CCN and INPs.*

The sentences have been re-phrased where appropriate as requested by the reviewer. Comparisons between the vertical profiles of a lidar system and coincident airborne in situ measurements have a long history. As mentioned already, the new aspects of our lidar-aircraft study are the comparison of cloud-relevant aerosol properties needed for CCN and INP profiles under heavy dust influence. Previous studies based on inversion of the lidar data often exclude dusty periods with highly depolarizing dust particles in their retrievals. The reviewer is right that the lidar does not observe CCN and INP, but rather derive or retrieve vertical profiles of CCN and INP number concentrations. The manuscript has been screened for such formulations.

*2) What is the basis for doing the CCN comparisons at 0.2% supersaturation? Is this a realistic supersaturation for clouds in this region? How would the comparison look for higher supersaturations (e.g., 0.4% or 0.6%)? For lower supersaturations (e.g., 0.1%)?*

Thank you for this remark. A supersaturation of 0.2% is a typical value (or standard) in CCN field observations and also a typical value describing usual supersaturations in the case of trade wind cumuli developing during fair weather conditions in the Caribbean. A sentence explaining our choice was added:

"The supersaturation of 0.2% with respect to water is motivated by the findings of Wex et al. (2016) as a typical value for trade wind cumuli in the Barbados region. Therefore, one of the CCN counters aboard the Falcon aircraft was set to this fixed supersaturation."

More details to the differences in the CCN concentrations for 0.4% and higher super saturation may be found in the paper of Mamouri and Ansmann (ACP, 2016).

*3) After reading the conclusions section, I'm unclear how this study advances the use of lidar observations to place constraints on CCN or INPs. The outlook that the authors lay out is that more comparisons in other environments are needed. Why? How will more comparisons be helpful? Would we expect the agreement between the lidar retrieval and in situ data to be better or the same? What contribution does this study make? I'd like to see more discussion that contextualizes how the present study is an advance upon the state-of-the-art.*

Motivated by the questions of the reviewer, the summary and conclusion section was completely rearranged and re-phrased. We summarize the main findings and draw main conclusions, and provide an outlook. We explain why we need more comparisons (aircraft vs lidar) and what our own next step will be (Cyprus campaign, aircraft vs lidar in an area with complex aerosol mixtures of fine-mode aerosol pollution and dust from the Middle East and Sahara).

To give more details here in the reply letter: It is essential to have a well-tested and validated approach to derive CCN and INP profiles from lidar measurements (and this means global coverage in the case of spaceborne lidars). As the reviewer already pointed out, the empirical relationships derived with long-term AERONET data need to be tested under different aerosol conditions. The present study focusses on mineral dust, but more inter comparisons, especially in continental polluted environments with complex mixtures of different aerosol types, are necessary. The good agreement in the $n_{250}$ number concentrations is already promising.

**Reply to Paul DeMott**

*General Comments*
*This paper continues a series of papers themed around the use of lidar retrievals to estimate CCN and INP profiles. These developments of capabilities are being followed with interest by a broad community since the applications and utility are obvious for regions where in situ data are not available, or not available with high frequency. This paper will make a nice incremental contribution to the growing literature base of this team, focusing here on comparison to aircraft data that did not have INP data to compare to. I do have some critical comments and suggestions in a few regards.*

*1) First, I believe that the nature of retrieval of multiple species contributions to aerosol number, mass, CCN and INP requires some additional description because this is a more recent development (versus retrievals from layers dominated by a single aerosol type) and so bears reiterating from its introduction over the last two years. If the Marinou et al. paper is accepted for publication, reference should be made to the detailing of the detailed schematic there.*
**Thank you for the hint. The description of the aerosol type separation has been improved and a reference to the recently published work by Marinou et al. (2019) has been made.**

*2) Secondly, I feel that the use of the DeMott et al. (2010) parameterization as specific to continental and non-dust contributions to INPs is not exactly correct, and this has implications. The "continental" definition in Mamouri and Ansmann (2016) neglects the fact that dust contributions to INPs were most certainly folded into the parameterization in a variety of environments. I see now that Marinou et al. (2018) have written, "As the majority of the samples used for D10 are non-desert continental aerosols, this INP parameterization has been considered to be suitable for addressing the immersion and condensation freezing activity of mixtures of anthropogenic haze, biomass burning smoke, biological particles, soil and road dust (Mamouri and Ansmann, 2016)." This is also a gross simplification, with the actual contributors unknown, and the likelihood that dust was folded in at a variety of levels of contribution. After all, one study was PACDEX, the Pacific Dust Experiment. Hence, strong caveats about potential duplication of INPs, and lack of assured attribution to all of the other types mentioned, are needed here. What one may really wish for are parameterizations for all relevant INP species instead. Substitution of D10 for the absence of such detailed information is not ideal, and so I am concerned that this is being glossed over. It is worrisome that this assumption seems to have propagated into a number of papers since 2016, and in some cases is even called "non-dust" or continental "pollution", the former not being true to the original paper and the latter being a true stretch in attribution that has never been supported by direct evidence.*
**Thank you, Paul DeMott, for correcting the use of your parameterization. We follow your comment and use the DeMott et al. (2010) parameterization now for all aerosols and not a specific type to not propagate the misinterpretation of your parameterization any further. As an input in the D10 parameterization we use the n250 derived for dust plus the n250 derived for continental or marine aerosol in the same layer as we derive n250 per aerosol type. In Fig. 4, now D10 is used for n250_dust+cont in the SAL (above 2 km height) and for n250_dust+marine in the marine aerosol layer below the SAL (dashed line). So we give all aerosol particles with radius larger than 250 nm as input in the D10 parameterization. For D15 we give only the dust particles as input.**

*3) I also wondered about the use of the groups' own parameterization of sea spray aerosols (based on DeMott et al., 2016, since that paper did not promote a direct parameterization) versus a marine-specific parameterization for the Atlantic region that is referenced in the introduction (McCluskey et al., 2018). Do they compare well? I obviously know the answer, but you might justify persisting with a parameterization that did*

*not as deeply consider "pure" marine as did the newer McCluskey paper. I realize that this is a very minor point, since marine INPs at -25 _C are minor contributors compared to mineral dusts in SAL conditions.*

**We clarified this point and implemented the McCluskey et al. (2018) parameterization for marine aerosol.**

*4) For the use of the DeMott et al. (2015) parameterization (D15), it seems that a decision has been made to not use the recommended 3x correction factor for immersion freezing that was justified in that paper? If so, the basis/reasoning for this should be stated.*

**Following your recommendation, we insert the correction factor of 3 in our calculations again. It is now explicitly stated in the manuscript. As we don't have in situ INP measurements to compare, we cannot state anything about the correction factor. The new parameterization by Harrison et al. (2019) for K-feldspar (1% for dust over Barbados) leads to results in the same range as D15 (with or without correction factor).**

*5) Finally, I think that it would be very useful to demonstrate retrievals in a profile that does not necessarily include dust or smoke overlying or mixing in the region above the marine boundary layer. That would represent the unperturbed case, and give insights into the behavior of the combined set of parameterizations when dust is not at all dominant.*

**Thank you for your suggestion. We included a new section 5.1 presenting a pure marine case without aerosol transport from Africa from the SALTRACE-2 winter campaign. This case shows the undisturbed marine background conditions of the Eastern Caribbean. The INP concentration calculated with the McCluskey (2018) parameterization is about 3 orders of magnitude lower than in the dusty summer cases using D15 parameterization.**

*Additional context to these comments and some additional specific questions/editorial comments for addressing before publication are listed below.*

***Specific Comments***

*1) Page 2, lines 2-4: What papers are you referring to in stating the implementation of these parameterization schemes? These are not all included in this present paper, although it would be interesting to see. Also, please note that there is no parameterization given in DeMott et al. (2016). This must have been created by the authors.*

**The reference to DeMott et al. (2016) was removed. We do not show all parameterizations mentioned in the introduction as the main focus of the paper is to evaluate the INP-relevant aerosol properties against in situ measurements and not to discuss the differences in all available INP parameterizations. This is a topic for its own, best with some in situ measured INP concentrations. For example, we calculated the INP concentrations in the SAL using the Ullrich et al. (2017) immersion freezing parameterization and found about one order of magnitude more INP.**

*2) Page 2, line 10: fix "several 10000 km" to state a range of distances expected.*

**Fixed. It is approximately 5000 km from the source to Barbados.**

*3) Page 2, lines 25-27: Note that as written, the sentence is repetitive in mentioning dust and smoke mixture at the beginning and ends of the sentence.*

**Corrected.**

*4) Page 3, lines 2: The continental aerosol designation is not mentioned here, as listed in Table 2. As stated above, this needs some serious caveats applied, namely that it is used in the absence of a true set of parameterizations that could describe other than mineral dust input, even though it definitely includes some influence from varied levels of mineral dusts in the studies used by D10. It was not intended to be specific or neglectful of any particular class of INPs.*

**The D10 parameterization is now used not for a specific type of aerosol, but rather for all aerosol particles with radius larger 250 nm as derived from our lidar observations. See comment above.**

*5) Last paragraph of Page 3, and start of Page 4: This discussion of assumptions on the hygroscopicity of mineral dust wanders some and never quite makes clear if kappa values for Saharan dust after transport to the region have been measured as low as is assumed or if this is an assumption based on the "fresh" nature of dust observed via say, microscopy studies. There is a difference, as trace amounts of materials can make a difference. In the end, it seems that the value selected of 0.02 is in the range of most measurements (i.e., not fully hydrophobic), and in the range estimated to be consistent with activation in clouds as submicron dust particles in the Eastern Atlantic (Twohy et al., GRL, 36, L01807, doi:10.1029/2008GL035846, 2009).*
**Thank you for your comment. We strengthened the discussion in this paragraph. The kappa values are not measured but taken from the literature assuming the "fresh" nature of dust even after long-range transport as support by several studies cited in the manuscript.**

*6) Page 4, line 14: The statement "The very hydrophilic sea salt particles (sodium chloride) have an activation diameter. . ." sounds awkward. Sea salt is hygroscopic. But sea salt is rarely the composition of sea spray particles alone, so why not say that "We assume a composition of sea salt for marine aerosols, and prescribe an activation diameter of. . ."*
**Thank you. The statement was rephrased.**

*7) Page 5, line 23: Perhaps discuss that cumuli attenuate the lidar, versus "disturbed" the measurements?*
**Changed.**

*8) Page 5, paragraph starting line 29: This is where I suggest that some elaboration on the methods for retrieving the contributions of different aerosols in a mixed scenario is given.*
**The description was improved here (and as well in Sect. 2.1):**
**"Because of the geographical location of Barbados, backward trajectories were not sufficient to decide whether the non-dust component was of marine or continental origin. Instead the method described in Ansmann et al. (2017) was applied which uses the fact that continental aerosol particles have a significantly higher lidar ratio (50 sr) due to considerable light absorption and much smaller particle sizes than the ones of marine aerosol particles (20 sr). The independently measured total particle extinction coefficient (from our Raman lidar measurements Ansmann et al., 1992) is compared to the sum of the extinction coefficients obtained by multiplying the type-separated backscatter coefficients with the respective type-dependent lidar ratios. An example will be shown in Section 5. A good agreement was found for continental pollution aerosol in the SAL and marine aerosol in the marine aerosol layer below."**

*9) Page 6, line 21: the CCN data from the Falcon are "measurements." They may have uncertainties, but they are not retrievals.*
**Corrected.**

*10) Page 7, first paragraph: Is the surface area used only for the marine parameterization? Do the dust parameterizations using s significantly differ from D15? I only wondered about the derivation of surface area if it was not going to be used.*
**Now the parameterizations by McCluskey et al. (2018) and Harrison et al. (2019) use s. Furthermore, the Ullrich et al. (2017) uses s. However, the U17 parameterization for immersion freezing tends to lead to quite high INP concentrations and is therefore not shown in the manuscript (but can be easily added).**

*11) Page 7, lines 8-10: This is a rather subjective statement about the likely role of the INP concentrations derived for the SAL. Clearly, direct cloud observations or cloud model simulations are likely needed to explore the implications, since such tropical cumuli are known to contain rather vigorous secondary ice formation processes through their deep supercooled layers (e.g., see Lasher-Trapp et al., J. Atmos. Sci., 73, 2547- 2564, 2016, and references to Lawson et al., 2015 and Heymsfield and Willis, 2014 therein).*
**The statement was removed.**

*12) Page 7, line 21: Consider replacing "Wrong in situ particle counting. . ." with uncertainties in in situ aerosol measurements. There is no support provided for how or why the measurements would be wrong. They are your only link to apparent ground truth.*
**The sentence was reformulated.**

*13) Page 8, line 15: suggest leading to "likely changes in trade wind cumulus cloud microphysical properties. . ." rather than "developments". Also, does it not depend on which layer dominates aerosol contributions to convective clouds?*
**We adopted your suggestion and reformulated the sentence.**

*14) Page 8, line 32: Suggest "reconciled" for "fixed"*
**Changed**.

*15) Page 9: The summary paragraph is a bit short in its outlook for the future. You would seem to benefit from more validation INP data, particularly for cases with and without dust, so the validity of the apparent knowledge of continental that you promote is also checked. And not only in dusty situations. Will you have INP data in any of the forthcoming campaigns? Quantifying other specific aerosol type contributions than dust and marine would appear useful as well.*
**The summary paragraph was completely reshaped. Airborne INP measurements would be indeed very helpful, but it is not easy to get them as most groups are measuring INP at ground level. So we have to stick to the comparison of the CCN and INP-relevant aerosol parameters. This is what we firstly have to validate from the lidar perspective. For the step from surface area and particle number concentration to INP concentrations we depend on the laboratory based INP parameterizations. To evaluate our algorithms under different aerosol situations is an important point for future activities. We will use measurements in the Eastern Mediterranean from the A-LIFE campaign (April 2017) where mixtures of urban pollution with Saharan and Middle Eastern dust as well as marine particles are observed.**

*16) Table 1 header: the cv coefficients need explanation. Are these the "conversion factors" mentioned?*
**They are the extinction-to-volume conversion factors now mentioned in the text.**

*17) Table 2: as mentioned, DeMott et al. (2016) does not include a parameterization, so that is not the appropriate reference for it. I suggest using the specific parameterization of McCluskey et al., if that is possible. Also the D15d reference needs to mention somewhere (if not in the table) what cf factor is used in this study.*
**Done.**

*18) Figure 4: I can note similar s values here as in DeMott et al. (2015) for the SAL over the Western Caribbean, but the predicted INP concentrations are a bit lower here at around -25_C. This motivated me to ask about the cf factor assumed for use in the parameterization in the present study. Also, below 2km, is it certain you are dealing with dust and not marine aerosols in all cases? Is this why the lidar profile showing higher surface area*

*and n250 on 22 June still leads to a decrease in D15-predicted INPs? Is that because you presume all of those particles are "continental"? This is where I think the application of a D10+D15 approach could lead to errors, and the only way to tell will be future in situ INP measurements.*

**You are right, below 2 km height we are dealing with a mixture of dust and marine particles. The plots were changed correspondently.**

*19) Figure 6 inspired me to ask what an unperturbed profile might look like, for example when there is not a strong dust or smoke or pollution layer over the clouds. Do you have any such data?*

**We included such case (26 Feb 2014) from the SALTRACE-2 winter campaign to show a case unperturbed by African aerosol. Unfortunately, the aircraft measurements are only available during the SALTRACE-1 campaign in summer 2013.**

[revised manuscript text omitted]